

# Exact full-RSB SAT/UNSAT transition in infinitely wide two-layer neural networks

**Brandon L. Annesi[1,2], Enrico M. Malatesta[1,3]⋆ and Francesco Zamponi[2]**

**1** Department of Computing Sciences, Bocconi University, 20136 Milano, Italy
**2** Dipartimento di Fisica, Sapienza Università di Roma, I-00185 Rome, Italy
**3** Institute for Data Science and Analytics, Bocconi University, 20136 Milano, Italy

⋆ enrico.malatesta@unibocconi.it

## Abstract

We analyze the problem of storing random pattern-label associations using two classes of continuous non-convex weights models, namely the perceptron with negative margin and an infinite-width two-layer neural network with non-overlapping receptive fields and generic activation function. Using a full-RSB Ansatz we compute the exact value of the SAT/UNSAT transition. Furthermore, in the case of the negative perceptron we show that the overlap distribution of typical states displays an overlap gap (a disconnected support) in certain regions of the phase diagram defined by the value of the margin and the density of patterns to be stored. This implies that some recent theorems that ensure convergence of Approximate Message Passing (AMP) based algorithms to capacity are not applicable. Finally, we show that Gradient Descent is not able to reach the maximal capacity, irrespectively of the presence of an overlap gap for typical states. This finding, similarly to what occurs in binary weight models, suggests that gradient-based algorithms are biased towards highly atypical states, whose inaccessibility determines the algorithmic threshold.

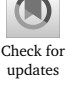

# 1  Introduction

One of the very first applications of the physics of disordered systems to machine learning has been the so-called storage problem. Given a model of a neural network, one asks what is the volume of networks in the space of weights which correctly classify a given instance of a (typically random) dataset. In a series of pioneering works [1–4], using tools previously

applied to the study of spin glasses, it was shown that in the limit of large system size a sharp SAT-UNSAT transition exists, where such volume goes to zero, as the ratio of the dataset-size to the data dimensionality is increased.

In recent years, this problem has seen a resurgence of interest, and has been framed both as a model of jamming and a model of machine learning. The first setting has been motivated by the realization that the storage problem of a simple one-layer neural network (called the *Perceptron* [5]) can be interpreted as the "simplest model of jamming" [6], and displays many of the critical properties of the jamming transition of soft matter systems [7]. Along these lines of research, some efforts have been devoted at understanding the universality class of this SAT-UNSAT (in this contest, *jamming*) transition [8,9], and identifying further models that belong to this same universality class [10,11].

The second setting, more relevant to the framing of this paper, has gained momentum as the need for a theoretical framework explaining the incredible success of deep learning has emerged. Indeed, most neural networks used in practice are so-called Interpolators, highly overparametrized networks which achieve zero error on the training set. Understanding how the set of these Interpolators behaves and how algorithms are able to find them has thus become crucial. Along these lines, several research directions have emerged. On the one hand, some efforts have been devoted at studying more realistic neural network and data models, including multiple layers, non-linear activation functions and non-i.i.d. data [12–18]. On the other hand, rather than asking questions about the existence and size of the set of solutions, its actual geometry has been investigated [19,20]. Simple properties such as the distance between solutions and their connectivity have proven to be insightful and a picture of how the high-dimensional loss landscape can have a profound impact on the behavior of algorithms has emerged [21,22].

In binary weights models, for example, the algorithmic threshold has been connected to the disappearance of a rare cluster of very dense solutions [22–24]. For continuous weight models instead, the picture is not as clear: the same tools used for binary models predict a threshold that can be easily overcome by simple algorithms [20].

In this work we consider two of these continuous models, the *Tree-Committee Machine* [10, 12,13,25] with arbitrary non-linearity and the *Spherical Negative Perceptron*, and settle a long-standing open problem about the numerical value of the SAT-UNSAT threshold. Previous estimates were all derived under the *Replica Symmetric* (RS) and *1-step Replica Symmetry Breaking* (1RSB) assumption, both of which only provide an approximation to the actual value.

Furthermore, we identify a new phase transition line between the *Full-Replica Symmetry Breaking* (fRSB) and the Gardner phase in the negative perceptron, where typical solutions develop a so called *Overlap Gap* [26]. We discuss this in connection to recently developed algorithms based on Approximate Message Passing [27,28], which provably finds solutions conditioned on the absence of this Overlap Gap.

The rest of the paper is organized as follows. In Section 2 we precisely define these models and the learning tasks we are interested in, namely the classification of random patterns and labels. In Section 3 we summarize the main steps in the analytical calculation we performed. In Section 4 we introduce a simple method through which we are able to compute the exact SAT/UNSAT threshold of those models with high precision. In Section 5 we study the transition to the Gardner phase starting from the fRSB phase, and propose an empirical method for the numerical estimation of this threshold. We also discuss where commonly used algorithms such as Gradient Descent are able to find solutions.

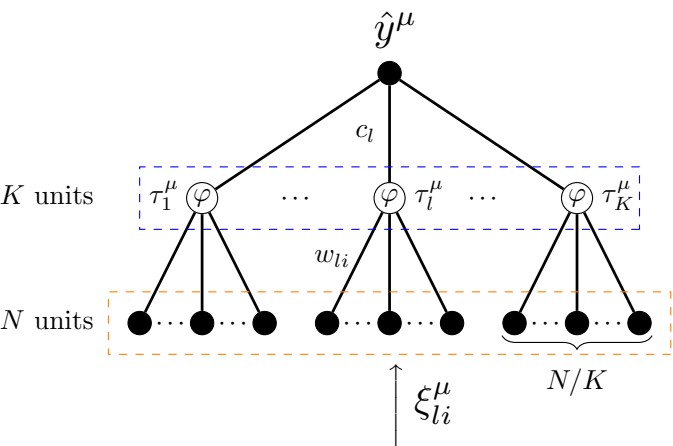

Figure 1: Tree committee machine architecture.

## 2 Model and learning task

The model that we will study in this work is a neural network with one hidden layer having non-overlapping receptive fields and fixed second layer weights, which is known in the statistical physics literature as the *tree-committee* machine. The architecture of the network is depicted in Fig. 1. Mathematically, given a $N$-dimensional input vector $\xi$, the output of the network is computed as

$$\hat{y} = \text{sign}\left( \frac{1}{\sqrt{K}} \sum_{l=1}^{K} c_l \, \varphi(\tau_l) \right), \tag{1}$$

where $K$ is the width of the hidden layer, $c_l$ are the weights of the second layer and $\tau_l$ is the $l$-th receptive field, given by

$$\tau_l^\mu \equiv \sqrt{\frac{K}{N}} \sum_{i=1}^{N/K} w_{li} \xi_{li} \,, \tag{2}$$

where $w_{li}$, $i \in \left[\frac{N}{K}\right]$, $l \in [K]$ are the $N$ weights of the first layer and $\varphi(\bullet)$ is a generic activation function; notice also that $N/K$ is an integer. We will consider in the following the case of *spherical weights*: individually $w_{li} \in \mathbb{R}$, but each branch of the weights is constrained to live on the $N/K$-dimensional sphere of radius $\sqrt{N/K}$,

$$\sum_{i=1}^{N/K} w_{li}^2 = \frac{N}{K} \,, \qquad l \in [K]. \tag{3}$$

The weights of the second layer will be considered fixed to $c_l = \pm 1$ or to $c_l = 1$ respectively depending if the activation function $\varphi$ is odd or not. Another choice could be to impose all the $c_l$ to be 1 and subtract a bias term $\sqrt{K}b$ inside the sign of equation (1), so that the preactivation of the output has zero mean. Notice that in the case of the identity activation function $\varphi(h) = h$ and for $K = 1$, we recover the *perceptron* architecture.

We are interested in learning the weights $\boldsymbol{w}$, in such a way that they correctly predict $P = \alpha N$ labels $y^\mu$, corresponding to $P$ input patterns $\xi^\mu$, $\mu \in [P]$. In the following we will call $\mathcal{D} = \left\{\xi^\mu, y^\mu\right\}_{\mu=1}^{P}$ the *training set* of the model. In the present paper we will be interested in the so-called *storage* problem, i.e. we will take inputs distributed as i.i.d. standard normal Gaussian variables $\xi_{li}^\mu \sim \mathcal{N}(0,1)$, $\forall \mu, i, l$ and the corresponding label will be $y^\mu = \pm 1$ with equal probability.

We are interested in classifying the input patterns in such a way that the preactivation of the output is aligned with the correct label within a certain margin $\kappa$; in other words we want the following constraints to be satisfied

$$\Delta^{\mu}(\boldsymbol{w};\kappa) \equiv \frac{y^{\mu}}{\sqrt{K}} \sum_{l=1}^{K} c_l \, \varphi(\tau_l^{\mu}) - \kappa \geq 0, \qquad \forall \mu \in [P]. \tag{4}$$

The quantity $\Delta^{\mu}(\boldsymbol{w};\kappa)$ is called the *stability* of the $\mu$-th pattern of the training set. We will call the set of $\boldsymbol{w}$ satisfying the constraints in equation (4) the *space of solutions* of the problem.

Since the labels are random, the problem is not always expected to be satisfiable (SAT). Indeed, in the large $N$ limit, the problem exhibit a *sharp* transition at a constrained density $\alpha_c(\kappa)$ above which the problem becomes unsatisfiable (UNSAT). In the following we will call $\alpha_c$ equivalently as *SAT/UNSAT transition* or *critical capacity*. In the soft matter literature, this also corresponds to the jamming transition [6].

Another interesting problem that we will analyze in the present paper is the so-called *negative perceptron* problem. This is recovered by taking $K = 1$, the identity activation $\phi(h) = h$ and a negative margin $\kappa < 0$. For simplicity, when $\varphi(\bullet)$ is a non-linear function, we will always focus on the case $\kappa = 0$.

## 2.1 Related works

Previous works on the tree-committee machine in the large width limit with sign [12, 13, 29] and other non-linear activation functions such as ReLU [15, 25] have only characterized the SAT/UNSAT transition in the Replica Symmetric (RS) or 1-step Replica Symmetry Breaking (1RSB) approximation. Recently, using fully-lifted random duality theory techniques, Refs. [30, 31] obtained results compatible with RS, 1RSB and 2-steps RSB approximations. One of the goals of the present work is to compute *exacly* $\alpha_c$ in the infinite-width limit regime.

The negative perceptron model has been recently studied in connection to jamming in high dimension [6, 8, 9, 32–35]: indeed patterns $\xi^{\mu}$, $\mu \in [P]$ can be thought to represent spherical obstacles to the possible position that a particle can occupy. The obstacles radius is determined by $\kappa$; jamming is attained by reaching the point where the particle has no available space and corresponds to the SAT/UNSAT transition. This can be achieved either by "inflating" the obstacles by increasing the margin, or by increasing the number of obstacles, i.e. $\alpha$. In this context the critical exponents of the model were computed [9] and were shown to be exactly the same as those observed in the jamming of spheres in large dimensions [8]. Recently, the tree-committee machine with several activation functions and the parity machine with a finite number of hidden units $K$ [10] have also been shown to pertain to the same universality class.

From the optimization point of view, imposing a negative margin is necessary in order to obtain a non-convex model: for $\kappa \geq 0$ indeed the space of solutions is convex and algorithms are able to reach capacity, which can be obtained exactly using an RS Ansatz [1, 3]. For $\kappa < 0$ the space of solution is instead non-convex [36] and, in the overparameterized regime $\alpha \ll 1$, it has been shown to be *star-shaped* [21]. From the point of view of algorithmic dynamics, at present it is difficult to compare the algorithmic threshold with the capacity transition since, as in the case of the committee machine, we only know approximations [20] or upper bounds [37] to the true value of the latter.

In [27], the authors develop an algorithm called incremental Approximate Message Passing (iAMP), originally devised in [28] for approximating the ground state of the Sherrington-Kirkpatrick model. Interestingly, this algorithm can be proven to reach capacity, provided that the typical states exhibit no overlap gap, i.e. the overlap distribution of typical states (equation (28)) has a connected support. This is what we refer in the rest of the paper as a no overlap gap condition (nOG).

Notice that the nOG condition is different from the no Overlap Gap Property (nOGP) introduced by Gamarnik [26] and that was connected to algorithmic hardness for stable algorithms. The OGP condition indeed requires that there exist $q_1$ and $q_2$ with $q_1 < q_2$ such that all pairs of solutions are at most at an overlap $q_1$ or at least an overlap $q_2$, i.e. there exists a "gap" since one cannot find solutions in the interval $q \in [q_1, q_2]$. As such, the OG condition we have introduced here is weaker with respect to the OGP since the last refers to all possible pairs of solutions, whereas the first one only to the typical ones. As a consequence the OGP implies the OG (while the nOG implies nOGP). A paradigmatic example showing the difference between OG and OGP is the binary perceptron: it has been shown that for any constraint density $\alpha > 0$ the problem exhibits OG (i.e. typical states are gapped) [38,39], but it is conjectured to display the OGP only above a critical value of the constrained density $\alpha > \alpha_{OGP} \simeq 0.77$ [22,40].

In the present paper we identify all the regions in the $(\kappa, \alpha)$ phase diagram that satisfy the nOG and we compute a new transition line separating a nOG from an overlap gapped phase for the typical states.

## 3 Accessing the entropy of solutions via the replica method

Following the seminal work by Gardner and Derrida [1,2] the volume of the space of solutions can be computed from the partition function

$$Z_{\mathcal{D}} = \int d\mu(\boldsymbol{w}) e^{-\beta \mathcal{L}(\boldsymbol{w})} = \int d\mu(\boldsymbol{w}) \prod_{\mu=1}^{P} e^{-\beta \ell(\Delta^{\mu}(\boldsymbol{w}; \kappa))}, \tag{5}$$

where the measure $d\mu(\boldsymbol{w})$ contains the spherical constraint in equation (3). The subscript $\mathcal{D}$ is there to remind that the partition function depends on the random realization of the dataset. Notice that depending on the loss function used, one may explore different kind of regions of the space of solutions; for example it has been shown that the cross-entropy loss in the large $\beta$ limit, tends to focus the measure $p(\boldsymbol{w}) \propto e^{-\beta \mathcal{L}(\boldsymbol{w})}$ over particular types of solutions having lower entropy, but more desirable properties such as a large robustness to perturbations over inputs and weights (flat minima) and low generalization error [19,20,40]. Here, we are particularly interested in studying the properties of the most probable solution that satisfies the constraints in equation (4); those *typical* solutions can be investigated by studying the *flat* measure over the set of all solutions. This corresponds to choosing the so called *error-counting* loss

$$\ell(x) \equiv \Theta(-x), \tag{6}$$

where $\Theta(x)$ is the Heaviside theta function, which is 1 if $x > 0$ and 0 otherwise. Using the error counting loss, the partition function in equation (5) becomes, in the large $\beta$ limit,

$$Z_{\mathcal{D}} = \int d\mu(\boldsymbol{w}) \prod_{\mu=1}^{P} \Theta(\Delta^{\mu}(\boldsymbol{w}; \kappa)) = \int d\mu(\boldsymbol{w}) \mathbb{X}_{\mathcal{D}}(\boldsymbol{w}; \kappa). \tag{7}$$

It is also called *Gardner volume* because it measures the volume of weights that satisfy the constraints of a correct classification of the training set, equation (4). We are interested in computing the average log-volume of solutions, i.e. the entropy of the system

$$\phi = \lim_{N \to \infty} \frac{1}{N} \overline{\ln Z_{\mathcal{D}}}, \tag{8}$$

where $\overline{\bullet}$ denotes the average over the disorder in the dataset.

Since the labels are random, we do not always expect the problem to be satisfiable (SAT). As shown in many previous works, in the large $N$ limit, the problem exhibit a *sharp* transition at a constrained density $\alpha_c(\kappa)$ above which the problem becomes unsatisfiable (UNSAT); at $\alpha_c$, correspondingly, the entropy diverges to $-\infty$. In the following we will call $\alpha_c$ equivalently as *SAT/UNSAT transition* or *critical capacity*. One of the goal of the present work is to compute *exactly* $\alpha_c$.

## 3.1 Replica method

Using the replica trick,

$$\overline{\ln Z_{\mathcal{D}}} = \lim_{n\to 0} \frac{\ln \overline{Z_{\mathcal{D}}^n}}{n}, \tag{9}$$

the average over the dataset can be performed considering $n$ as an integer. In the derivation, the order parameters

$$q_l^{ab} \equiv \frac{K}{N} \sum_{i=1}^{N/K} w_{li}^a w_{li}^b, \qquad a < b \in [n], \quad l \in [K], \tag{10}$$

naturally appear. They represent the overlap between the same hidden unit of two independent replicas of the systems. The overlap between different hidden units does not contribute because they are connected to non-overlapping, uncorrelated portion of the input. We enforce the definition (10) by using delta functions and their integral representations; this will in turn introduce the conjugated parameters $\hat{q}_l^{ab}$ with $a \leq b \in [n]$, $l \in [K]$. Notice that we need also the diagonal conjugated overlaps $\hat{q}_l^{aa}$ in order to enforce the spherical constraint in equation (3). In the end we get the following representation of the averaged replicated partition, function

$$\overline{Z_{\mathcal{D}}^n} = \int \prod_{\substack{a<b \\ l}} dq_l^{ab} \prod_{\substack{a\leq b \\ l}} d\hat{q}_l^{ab} \, e^{NS(q,\hat{q})}, \tag{11}$$

where we have defined

$$S(q, \hat{q}) \equiv G_S(q, \hat{q}) + \alpha G_E(q), \tag{12a}$$

$$G_S(q, \hat{q}) \equiv \frac{1}{2K} \sum_{ab} \sum_l q_l^{ab} \hat{q}_l^{ab} - \frac{1}{2K} \sum_{l=1}^K \ln \det \hat{q}_l, \tag{12b}$$

$$G_E(q) \equiv \ln \mathbb{E}_y \int \prod_{la} \frac{d\lambda_l^a d\hat{\lambda}_l^a}{2\pi} \prod_a e^{-\beta \ell \left( \frac{y}{\sqrt{K}} \sum_{l=1}^K c_l \, \varphi(\lambda_l^a) - \kappa \right)} e^{i \sum_{la} \lambda_l^a \hat{\lambda}_l^a - \frac{1}{2} \sum_{ab,l} q_l^{ab} \hat{\lambda}_l^a \hat{\lambda}_l^b}, \tag{12c}$$

and we have understood that $q_l^{aa} = 1$ because of the spherical constraint (3). The conjugated parameters satisfy saddle point equations that can be explicitly solved: $q_l^{ab} = \left[ \hat{q}_l^{-1} \right]^{ab}$. Therefore the averaged replicated partition function can be written more compactly as

$$\overline{Z_{\mathcal{D}}^n} = \int \prod_{\substack{a<b \\ l}} dq_l^{ab} \, e^{NS(q)}, \tag{13}$$

where

$$S(\boldsymbol{q}) \equiv G_S(\boldsymbol{q}) + \alpha G_E(\boldsymbol{q}), \tag{14a}$$

$$G_S(\boldsymbol{q}) \equiv \frac{1}{2K} \sum_{l=1}^{K} \ln \det \boldsymbol{q}_l, \tag{14b}$$

$$G_E(\boldsymbol{q}) \equiv \ln \mathbb{E}_y \int \prod_{la} \frac{d\lambda_l^a d\hat{\lambda}_l^a}{2\pi} \prod_a e^{-\beta\ell\left(\frac{y}{\sqrt{K}}\sum_{l=1}^{K} c_l \varphi(\lambda_l^a) - \kappa\right)} e^{i\sum_{la}\lambda_l^a\hat{\lambda}_l^a - \frac{1}{2}\sum_{ab,l} q_l^{ab}\hat{\lambda}_l^a\hat{\lambda}_l^b}. \tag{14c}$$

Notice that we recover the perceptron by imposing $\varphi(x) = x$ and $K = 1$. We write both the entropic and energetic terms for later convenience

$$G_S(\boldsymbol{q}) = \frac{1}{2} \ln \det \boldsymbol{q}, \tag{15a}$$

$$G_E(\boldsymbol{q}) \equiv \ln \mathbb{E}_y \int \prod_a \frac{d\lambda^a d\hat{\lambda}^a}{2\pi} \prod_a e^{-\beta\ell(y\lambda^a - \kappa)} e^{i\sum_a \lambda^a\hat{\lambda}^a - \frac{1}{2}\sum_{ab} q^{ab}\hat{\lambda}^a\hat{\lambda}^b} \tag{15b}$$

$$= \ln \mathbb{E}_y \, e^{\frac{1}{2}\sum_{ab} q^{ab} \frac{\partial^2}{\partial h^a \partial h^b}} \prod_a e^{-\beta\ell(yh^a - \kappa)} \bigg|_{h^a=0}.$$

In the last step we have integrated over the $\hat{\lambda}^a$ variables and used the following set of identities

$$\int \prod_a \frac{d\lambda^a}{\sqrt{2\pi \det \boldsymbol{q}}} e^{-\frac{1}{2}\sum_{ab}[\boldsymbol{q}^{-1}]^{ab}\hat{\lambda}^a\hat{\lambda}^b} \prod_a g(\lambda^a) = \int \prod_a D\lambda^a \prod_a g\left(\sum_b [\sqrt{\boldsymbol{q}}]^{ab} \lambda^b\right)$$

$$= \int \prod_a D\lambda^a \, e^{\sum_{ab}[\sqrt{\boldsymbol{q}}]^{ab} \lambda^b \frac{d}{dh_a}} \prod_a g(h_a) \bigg|_{h_a=0}$$

$$= e^{\frac{1}{2}\sum_{ab} q^{ab} \frac{d^2}{dh_a dh_b}} \prod_a g(h_a) \bigg|_{h_a=0}, \tag{16}$$

where $g(\bullet)$ is a generic function and $D\lambda \equiv \frac{d\lambda}{\sqrt{2\pi}} e^{-\frac{\lambda^2}{2}}$. We have also used the notation $\sqrt{\boldsymbol{q}}$ to denote the square root of the symmetric (and therefore positive semidefinite) overlap matrix $q^{ab}$.

## 3.2 Large width limit

The large number of hidden units limit can be performed before imposing the Ansatz over the replica indices of the overlap matrix $q_l^{ab}$. It is important to note however that we perform the limit $K \to \infty$ after the limits $P, N \to \infty$. Indeed we are implicitly implying that the $G_S$ and $G_E$ functions are calculated on the saddle points first, and only then we consider their large $K$ limit. The number of hidden units, although infinite, is of smaller order compared to both $N$ and $P$. Another important observation is that since the weights are not overlapping and have access to uncorrelated portions of the input, clearly $q_l^{ab}$ must be independent on $l$ on average. We can exploit this to simplify notably the entropic and energetic terms. The entropic term is easy and it reads

$$G_S(\boldsymbol{q}) = \frac{1}{2} \ln \det \boldsymbol{q}. \tag{17}$$

In the energetic term (14c) we have instead to use the central limit theorem on the variable $u_a = \frac{1}{\sqrt{K}} \sum_{l=1}^{K} c_l \varphi(\lambda_l^a)$. This can be done extracting the variables $u_a$ from the loss function in (14c) via $n$ delta functions, inserting their integral representations, Taylor expanding at

second order and re-exponentiating. We show explicitly those steps in Appendix B.2.1; we find

$$G_E(\boldsymbol{q}) = \ln \mathbb{E}_y \int \prod_a \frac{du_a d\hat{u}_a}{2\pi} e^{i\sum_a u_a \hat{u}_a} \prod_a e^{-\beta\ell(yu_a-\kappa)} e^{-i\sum_a \hat{u}_a M_a - \frac{1}{2}\sum_{ab}\Delta_{ab}\hat{u}_a\hat{u}_b}, \tag{18}$$

where $M_a$ and $\Delta_{ab}$ represents respectively the mean and the covariance matrix of the variable $u_a$, i.e.

$$M_a \equiv m_c \int \prod_a \frac{d\lambda_l^a d\hat{\lambda}_l^a}{2\pi} e^{i\sum_a \lambda_l^a \hat{\lambda}_l^a - \frac{1}{2}\sum_{ab}q^{ab}\hat{\lambda}_l^a\hat{\lambda}_l^b}\varphi(\lambda_l^a) \equiv m_c \left\langle \varphi(\lambda^a)\right\rangle, \tag{19a}$$

$$\Delta_{ab} \equiv \sigma_c \left[ \left\langle \varphi(\lambda^a)\varphi(\lambda^b)\right\rangle - \left\langle \varphi(\lambda^a)\right\rangle\left\langle \varphi(\lambda^b)\right\rangle \right], \tag{19b}$$

with $m_c \equiv \frac{1}{\sqrt{K}}\sum_{l=1}^K c_l$ and $\sigma_c \equiv \frac{1}{K}\sum_{l=1}^K c_l^2$. Using the identity in equation (16) the energetic term, $M_a$ and $\Delta_{ab}$ can all be compactly written in the following form

$$G_E(\boldsymbol{q}) = \ln \mathbb{E}_y \; e^{\frac{1}{2}\sum_{ab}\Delta_{ab}\frac{\partial^2}{\partial h_a \partial h_b}} \prod_a e^{-\beta\ell(y(M_a+h_a)-\kappa)} \Bigg|_{h_a=0}, \tag{20a}$$

$$M_a \equiv m_c \; e^{\frac{1}{2}\sum_{ab}q^{ab}\frac{\partial^2}{\partial s_a \partial s_b}} \varphi(s_a)\Bigg|_{s_a=0}, \tag{20b}$$

$$\Delta_{ab} \equiv \sigma_m \; e^{\frac{1}{2}\sum_{ab}q^{ab}\frac{\partial^2}{\partial s_a \partial s_b}} \varphi(s_a)\varphi(s_b)\Bigg|_{s_a=0} - m_c^2 M_a M_b. \tag{20c}$$

Notice that in our case the mean $M_a$ is always vanishing: if the activation function is odd indeed $\langle\varphi(\lambda^a)\rangle = 0$, whereas if the activation function is even $m_c = 0$ since $c_l = \pm$ with equal probability in order to prevent the model to have a bias towards positive or negative labels. We therefore get the following integral representation of the model in the large $K$ limit:

$$\overline{Z_{\mathcal{D}}^n} = \int \prod_{a<b} dq^{ab} \; e^{NS(\boldsymbol{q})}, \tag{21a}$$

$$S(\boldsymbol{q}) = \frac{1}{2}\ln\det\boldsymbol{q} + \alpha \ln\left(\mathbb{E}_y \; e^{\frac{1}{2}\sum_{ab}\Delta_{ab}\frac{\partial^2}{\partial h_a \partial h_b}} \prod_a e^{-\beta\ell(yh_a-\kappa)}\Bigg|_{h_a=0}\right). \tag{21b}$$

Notice that this expression is exactly equal in form to that of the perceptron model, see equation (15); the only difference is that instead of having the matrix $q^{ab}$ we have an effective order parameter $\Delta_{ab}$ which is a function through $\varphi(\cdot)$ of $q^{ab}$. This has been evidenced for the first time in [25]. The quantity $\Delta_{ab}$ is also exactly identical to the so-called Neural Network Gaussian Process (NNGP) kernel [41] that appears as the covariance matrix of the function implemented by a neural network at initialization (i.e. with random weights) in the infinite width limit and given two different inputs [42, 43]. Here, the only difference is that this quantity does not depend on the overlap between those two inputs, but it depends instead on the average overlap $q^{ab}$ between two different replicas of the weights extracted from the Gibbs measure.

### 3.2.1 Saddle point equations

In the large $N$ limit, the averaged replicated partition function in equation (21) is dominated by the saddle points of the action $S(\boldsymbol{q})$. The entropy of the system can be therefore written as

$$\phi = \lim_{n\to 0}\max_{\boldsymbol{q}} \frac{S(\boldsymbol{q})}{n}. \tag{22}$$

The stationary points of the action can be obtained by imposing that the first derivative of the action vanishes. This set of $\frac{n(n-1)}{2}$ saddle point equations read, in the large width limit, as

$$q_{cd}^{-1} = -\alpha \frac{d\Delta_{cd}}{dq_{cd}} \frac{\mathbb{E}_y \, e^{\frac{1}{2}\sum_{ab}\Delta_{ab}\frac{\partial^2}{\partial h_a \partial h_b}} \frac{\partial^2}{\partial h_c \partial h_d} \prod_a e^{-\beta \ell(yh_a-\kappa)}\Big|_{h_a=0}}{\mathbb{E}_y \, e^{\frac{1}{2}\sum_{ab}\Delta_{ab}\frac{\partial^2}{\partial h_a \partial h_b}} \prod_a e^{-\beta \ell(yh_a-\kappa)}\Big|_{h_a=0}}, \qquad c < d \in [n], \qquad (23)$$

where

$$\frac{d\Delta_{ab}}{dq^{ab}} = e^{\frac{1}{2}\sum_{cd} q^{cd}\frac{\partial^2}{\partial s_c \partial s_d}} \frac{\partial \varphi(s_a)}{\partial s_a} \frac{\partial \varphi(s_b)}{\partial s_b}\Big|_{s_a=0}. \qquad (24)$$

### 3.3 Full replica symmetry breaking Ansatz and variational formulation

In order to solve the saddle point equations in the small $n$ limit, one needs to impose some type of Ansatz on the structure of the replica overlap matrix $q^{ab}$. Here we consider the most general type of Ansatz, the $k$-steps Replica Symmetry Breaking ($k$-RSB) Ansatz [44–46], in which it is assumed that the overlap matrix assumes the $k + 1$ values $q_0, q_1, \ldots, q_k$. Defining the set of integers $1 = m_k \le m_{k-1} \le \cdots \le m_0 \le m_{-1} \equiv n$ with $m_{s-1}$ divisible by $m_s$ for $s = 0, \ldots, k-1$ the overlap matrix $q^{ab}$ is written in the $k$-RSB Ansatz as

$$q^{ab} = q_0 + \sum_{s=0}^{k} (q_{s+1} - q_s) I_{n,m_s}^{ab}, \qquad (25)$$

where $I_{n,m_s}^{ab}$ is the $(a, b)$ element of a block matrix of size $n \times n$ whose blocks have size $m_s \times m_s$ and contains all ones and all zeros respectively inside and outside the blocks. We have understood in the previous equation that $q_{k+1} = 1$.

In the following we will use the square bracket notation $[\bullet]_s$ to denote the operation of extracting step $s + 1$ from the $k$-step RSB matrix in its argument, i.e., for example, $\left[q^{ab}\right]_s = q_s$. As we show in appendix B also the NNGP kernel $\Delta_{ab}$ assumes a $k$-RSB form with the same block structure of $q^{ab}$; in addition the $s + 1$-th step of $\Delta_{ab}$ is given by a simple function of the $(s + 1)$-th step of the matrix $q^{ab}$

$$[\Delta_{ab}]_s = \int Dx \left[\int Dy \, \varphi\left(\sqrt{q_s}\,x + \sqrt{1-q_s}\,y\right)\right]^2 \equiv \Delta(q_s). \qquad (26)$$

We report in appendix B the expression of the entropic and energetic term in the small $n$-limit for the $k$-step RSB Ansatz.

In the small $n$ limit, the parameterization (25) is equivalent to requiring that the matrix $q^{ab}$ is parameterized by a stepwise function $q(x)$ in the interval $x \in [0, 1]$

$$q(x) = q_s, \qquad x \in [m_{s-1}, m_s), \qquad s = 0, \ldots, k. \qquad (27)$$

In the large number of steps limit $q(x)$ tends to a continuous function and so does the NNGP kernel function $\Delta(q)$. This is what is called full-RSB Ansatz (fRSB). When we are in the fRSB phase, we expect the $q(x)$ that maximises the free energy [21] to have the following shape: for $x \in [0, x_m)$ and $x \in [x_M, 1)$, $q(x)$ is constant and equal to $q_m$ and $q_M$ respectively, while for $x \in [x_m, x_M]$ it is a continuous monotonic function of $x$. In the Replica Symmetric (RS) phase instead, we expect the $q(x)$ to be constant and equal to a single value $q$.

Although the function $q(x)$ is not of easy interpretation, it is connected to a fundamental quantity, namely the probability distribution of the overlap between two samples of the

uniform measure over solutions

$$P(q) = \overline{\int d\mu(\boldsymbol{w}^1) d\mu(\boldsymbol{w}^2) \mathbb{X}_{\mathcal{D}}(\boldsymbol{w}^1;\kappa) \mathbb{X}_{\mathcal{D}}(\boldsymbol{w}^2;\kappa) \delta\left(q - \frac{K}{N}\sum_{i=1}^{N/K} w_{li}^1 w_{li}^2\right)}. \tag{28}$$

Indeed it can be shown that if we denote by $x(q)$ the inverse function of $q(x)$, then $P(q) = \frac{dx(q)}{dq}$ (or in other word $x(q)$ is the CDF of $P(q)$).

Performing the continuous limit, the fRSB entropy can be written as

$$\phi = \mathcal{G}_S + \alpha \mathcal{G}_E, \tag{29a}$$

$$\mathcal{G}_S \equiv \lim_{n\to 0} \frac{G_S}{n} = \frac{1}{2}\left[\ln(1-q_M) + \frac{q_m}{\lambda_m} + \int_{q_m}^{q_M} \frac{dq}{\lambda(q)}\right], \tag{29b}$$

$$\mathcal{G}_E \equiv \lim_{n\to 0} \frac{G_E}{n} = \int dh \, \mathcal{N}_{\Delta(q_m)-\Delta(0)}(h) f(q_m, h), \tag{29c}$$

having indicated with $\mathcal{N}_\sigma(h) \equiv \frac{e^{-\frac{h^2}{2\sigma}}}{\sqrt{2\pi\sigma}}$ and by $\lambda(q)$ the continuous limit of the eigenvalues of a $k$-RSB matrix (see also section A.2), i.e.

$$\lambda(q) = \int_q^1 dq' x(q'). \tag{30}$$

The function of two variables $f$ in the energetic term satisfies the following partial differential equation (PDE) [47, 48]

$$f(q_M, h) = \ln \int dz \, \mathcal{N}_{\Delta(1)-\Delta(q_M)}(z+h) e^{-\beta \ell(z-\kappa)}, \tag{31a}$$

$$\dot{f}(q,h) = -\frac{1}{2}\dot{\Delta}(q)\left[f''(q,h) + x(q)f'(q,h)^2\right], \tag{31b}$$

having denoted with a upper dot the derivative with respect to $q$ and with a prime the derivative with respect to $h$. The second equation (31b) is a slight variation to the Parisi's equation which is obtained in the case $\Delta(q) = 1$ i.e. in the linear activation (perceptron) case. Notice that for both the error counting loss and the quadratic hinge loss, the initial condition, equation (31a), can be explicitly solved analytically; in particular, in the large $\beta$ limit, in both cases one has

$$f(q_M, h) = \ln H\left(\frac{\kappa + h}{\sqrt{\Delta(1)-\Delta(q_M)}}\right), \tag{32}$$

where $H(x) \equiv \frac{1}{2}\text{Erfc}\left(\frac{x}{\sqrt{2}}\right)$. The saddle point equations in the continuous limit are difficult to derive differentiating equation (29c) with respect to $x(q)$, because $f$ depends implicitly on $x(q)$ through equation (31b). As suggested in [49] we can remove this dependence by using Lagrange's method [48, 49]

$$\begin{aligned} \phi_{\text{var}} = \frac{1}{2}&\left[\ln(1-q_M) + \frac{q_m}{\lambda_m} + \int_{q_m}^{q_M}\frac{dq}{\lambda(q)}\right] + \alpha\int dz\,\mathcal{N}_{\Delta(q_m)-\Delta(0)}(z)f(q_m,z) \\ &- \alpha\int_{-\infty}^{+\infty} dh\,P(q_M,h)\left[f(q_M,h) - \ln\int dz\,\mathcal{N}_{\Delta(1)-\Delta(q_M)}(z+h)e^{-\beta\ell(z-\kappa)}\right] \\ &+ \alpha\int_0^1 dq\int_{-\infty}^{+\infty} dh\,P(q,h)\left[\dot{f}(q,h) + \frac{\dot{\Delta}(q)}{2}\left(f''(q,h) + x(q)f'(q,h)^2\right)\right]. \end{aligned} \tag{33}$$

Deriving $\phi_{\text{var}}$ with respect to $x(q)$ we get the saddle point equations in the continuous limit

$$\frac{q_m}{\lambda_m^2} + \int_{q_m}^{q} \frac{dp}{\lambda^2(p)} = \alpha\dot\Delta(q)\int dh\, P(q,h)f'(q,h)^2. \tag{34}$$

Differentiating with respect to $f(q_m,h)$ and $f(q,h)$ we get that the function $P$ satisfies a PDE of the Fokker-Planck type

$$P(q_m,h) = \mathcal{N}'_{\Delta(q_m)-\Delta(0)}(h), \tag{35a}$$

$$\dot P(q,h) = \frac{\dot\Delta(q)}{2}\left[P''(q,h) - 2x(q)\big(P(q,h)f'(q,h)\big)'\right], \tag{35b}$$

which can be shown to be equal to the continuous limit of iteration rule given in appendix, equation (B.29).

We show in appendix B how to solve equations (31) and (34) numerically by writing them in a discretized version that correspond to a finite number $k$ of steps of RSB. Once they are solved for a particular guessed value of $q(x)$ in the interval $[x_m, x_M]$, the updated $q(x)$ can be computed from equation (34).

## 3.4 Instability of the Ansatz

The continuous limit and the variational formulation of the saddle point described above can be also useful as a tool to derive equations describing the instability of the Ansatz itself. In order to do that we need to derive equation (34) written in terms of $x$,

$$\frac{q_m}{\lambda_m^2} + \int_{x_m}^{x} dy\, \frac{\dot q(y)}{\lambda^2(y)} = \alpha\dot\Delta(q(x))\int dh\, P(x,h)f'(x,h)^2, \tag{36}$$

with respect to $x$. We use the identity

$$\frac{\partial}{\partial x}\int dh\, P(x,h)g(x,h) = \int dh\, P(x,h)\Omega(x,h)g(x,h), \tag{37}$$

where $\Omega(x,h)$ is the differential operator

$$\Omega(x,h) = \frac{\partial}{\partial x} + \frac{\dot\Delta}{2}\frac{dq}{dx}\left(\frac{\partial^2}{\partial h^2} + 2xf'(x,h)\frac{\partial}{\partial h}\right). \tag{38}$$

Deriving equation (36) with respect to $x$ once, *assuming that* $\frac{dq}{dx} \neq 0$ (i.e. $x$ is considered to be in the interval $[x_m, x_M]$) and using Parisi's equation (31) we have

$$\frac{1}{\lambda^2(q)} = \alpha\ddot\Delta(q)\int dh\, P(q,h)f'(q,h)^2 + \alpha\dot\Delta^2(q)\int dh\, P(q,h)f''(q,h)^2. \tag{39}$$

This equation computed at the $k$-RSB level will give us a prediction of the Ansatz instability, i.e. the value of $\alpha$ for which the chosen Ansatz does not hold anymore. In the appendix we show how this expression reproduces two transition lines, the de Almeida-Thouless (dAT) instability [50], which signals a transition from a stable RS to a full-RSB phase, and the so-called Gardner transition line [51], which signals a transition from a stable 1RSB phase to a Gardner phase which we define in Section 5.1. Indeed the first is obtained by evaluating equation (39) with a Replica Symmetric (RS) Ansatz ($q(x) = q$ for any $x \in [0,1]$), while the second by evaluating it using a one-step RSB Ansatz. We show later, in the case of the negative perceptron a plot of the dAT line and the Gardner transition line.

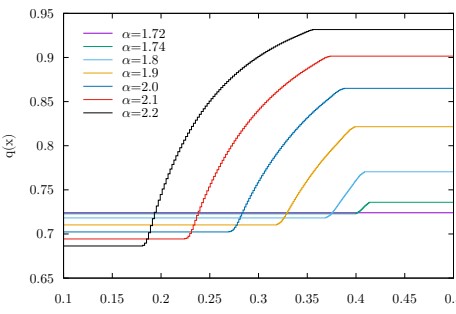 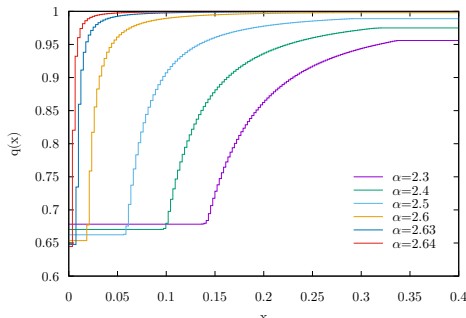

Figure 2: Overlap $q(x)$ for the infinite-width tree-committee machine, with ReLU non-linearity near the onset of RSB which happens at $\alpha_{\text{dAT}} \sim 1.7212$ [25] (left panel), and near the critical capacity regime (right panel).

### 3.5 Breaking point update

From the numerical point of view, even the breaking points $x_m$ and $x_M$ need to be found. An update equation for each one of them can be obtained [52] deriving equation (39) with respect to $x$. Again assuming $\frac{dq}{dx} \neq 0$ and solving for $x$

$$x = \frac{\lambda(x)}{2} \frac{\int dh\, P(x,h) \left[ \dddot{\Delta} f'(x,h)^2 + 3\dot{\Delta}\ddot{\Delta} f''(x,h)^2 + \dot{\Delta}^3 f'''(x,h)^2 \right]}{\int dh\, P(x,h) \left[ \ddot{\Delta} f'(x,h)^2 + \dot{\Delta}^2 f''(x,h)^2 + \lambda(x)\dot{\Delta}^3 f''(x,h)^3 \right]}, \tag{40}$$

which in the case of the identity activation function $\Delta(q) = q$ reduces to [9]

$$x = \frac{\lambda(x)}{2} \frac{\int dh\, P(x,h) f'''(x,h)^2}{\int dh\, P(x,h) [f''(x,h)^2 + \lambda(x) f''(x,h)^3]}. \tag{41}$$

Once equations (31), (35) and (34) are solved for a guess of $x_m$ and $x_M$, they can be updated using equation (40); the whole process is iterated until convergence is reached. We refer to the appendix B for an in-depth discussion of the numerical procedure used.

In Fig. 2 we show the resulting plots of $q(x)$ for several values of $\alpha$ starting from the onset of RSB at $\alpha_{\text{dAT}}$ in the case of the ReLU activation function $\text{ReLU}(z) = \max(0, z)$.

## 4 Exact determination of the SAT/UNSAT transition

In order to determine the SAT/UNSAT transition, a possible strategy is to perform the $q_M \to 1$ limit inside the fRSB equations. This has been performed in [8, 9], in order to determine the critical exponents of jamming. However the resulting equations are not easy to analyze numerically. Here we adopt another simpler approach that consists in evaluating an observable whose behavior near the SAT/UNSAT transition can be analytically predicted.

This observable is called the *reduced pressure* and it is proportional to the derivative of the free entropy with respect to the margin

$$\tilde{p} = -\frac{1}{\alpha} \frac{\partial \phi}{\partial \kappa}. \tag{42}$$

The name "pressure" comes from the fact that when we differentiate the free energy with respect to the volume one gets the pressure: in the sphere packing interpretation of the negative perceptron problem, a variation with respect to $\kappa$ is indeed equivalent to a change of

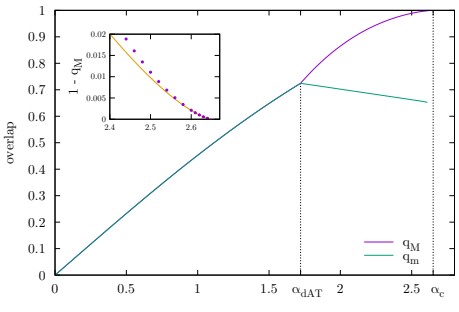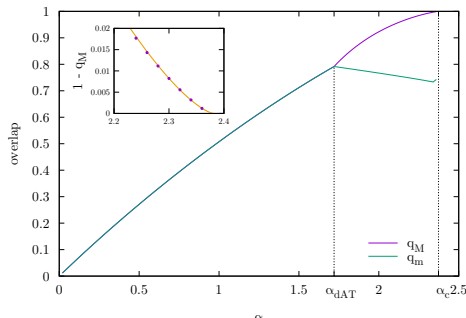

Figure 3: Minimum and maximal overlap $q_m$ and $q_M$ as a function of $\alpha$ in the case of the ReLU (left panel) and Erf activation functions (right) with $\kappa = 0$. For $\alpha \leq \alpha_{dAT}$, the RS Ansatz is correct so $q_m = q_M$. For $\alpha \to \alpha_c$ we have that $q_M \to 1$. (Inset) We show that $q_M$ scales as a power law, see equation (44), with an exponent $\sigma \simeq 1.4157$. Dots are exact numerical solutions, lines are power-law fits.

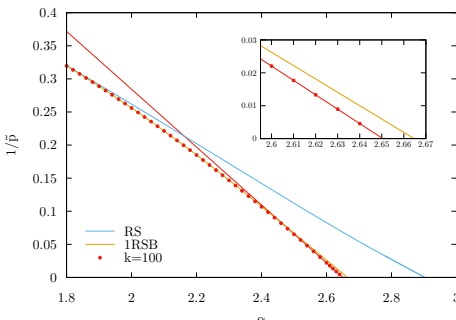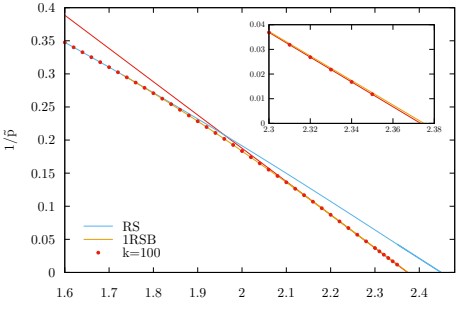

Figure 4: Inverse reduced pressure as a function of the constraint density $\alpha$ in the case of the infinite-width tree-committee machine, with ReLU (left panel) and Erf (right) activation functions with $\kappa = 0$. The blue and orange lines represents RS and 1RSB predictions. The red dots represent the solutions obtained by using $k = 100$ steps of RSB. For $\alpha \to \alpha_c$ the inverse reduced pressure scales as $\tilde{p}^{-1} \sim \alpha - \alpha_c$. The red line represents a fit to the $k$-RSB data near the critical capacity.

the particle volume [6]. We refer the reader to appendix C for a connection of the reduced pressure to the stability distribution. Reminding that the evolution equations for the functions $\tilde{f}(q_m, h) = f(x_m, -h - \kappa)$ and $P$ are independent on $\kappa$, one gets

$$\tilde{p} = -\frac{1}{\alpha}\frac{\partial \phi}{\partial \kappa} = -\int dh\, P(q_m, h) f'(x_m, h) = -\int dh\, P(q_M, h) f'(q_M, h). \tag{43}$$

Now we use the (not yet mathematically proven) fact that upon approaching the SAT/UNSAT transition, $\tilde{p}$ scales as [9,53]

$$\tilde{p} \propto \frac{1}{\alpha_c - \alpha}. \tag{44}$$

We show in Fig. 4 a validation of this scaling from the numerical solution of our fRSB equation. We applied this strategy to the non-linear two-layer networks defined in section 2 with zero margin, $\kappa = 0$. We show in Fig. 4 the inverse reduced pressure as a function of $\alpha$ for the ReLU and Erf activations computed using $k = 100$ steps of RSB; a linear fit to the numerical data is also presented. We show for comparison also the inverse reduced pressure computed at the RS and 1RSB level. In Table (1) we summarize our findings for the value of the SAT/UNSAT

Table 1: dAT and exact SAT/UNSAT transition for some activation functions with $\kappa = 0$. We also show for comparison the SAT/UNSAT transition computed in the RS approximation.

|  | ReLU | Tanh | Erf | Swish |
|---|---|---|---|---|
| $\alpha_{\mathrm{dAT}}$ | 1.721195 | 1.7530 | 1.71995 | 1.805634 |
| $\alpha_c^{\mathrm{RS}}$ | $\frac{2\pi}{\pi-1} \simeq 2.934$ | 2.3556 | 2.4514 | 2.42416 |
| $\alpha_c^{\mathrm{1RSB}}$ | 2.66428 | 2.306265 | 2.37499 | 2.3855699 |
| $\alpha_c^{\mathrm{fRSB}}$ | 2.6504(5) | 2.3049(0) | 2.3733(5) | 2.3838(3) |

transition for several activation functions. We also report the constraint density where RSB effects arise and the SAT/UNSAT transition computed in the RS and 1RSB approximations (whose derivations can be found respectively in appendix B.4 and B.5).

# 5 Gardner phase in the negative perceptron and the no overlap gap condition

In this section we focus on the case of the Negative Perceptron. While in two-layer networks a non-convexity is already present due to the non-linear activation function of the hidden layer, in the case of the perceptron one needs to achieve non-convexity by using a negative margin $\kappa < 0$. We will thus be concerned with the whole $(\kappa, \alpha)$ phase diagram, while in the previous section we limited ourselves to the $\kappa = 0$ case. In subsection 5.1 we remind the full phase diagram of the model, whereas in subsection 5.2 we unveil the presence of a line separating two phases, where typical states respectively have or do not have an overlap gap. We refer to appendix B.5 the phase diagram of the tree-committe machine with ReLU activation.

## 5.1 Phase diagram

Depending on the value of the load $\alpha$ and the margin $\kappa$, the model exhibits a variety of phases, the boundaries of which were calculated in [9]. In the appendix we sketch how these lines can be estimated, while here we summarise what the phases are, and what type of $q(x)$ we expect in each phase. A plot of the phase diagram is reported in Figure 5.

For $\alpha < \alpha_{dAT}$, the RS solution is stable, and we thus expect $q(x)$ to be constant. Increasing $\alpha$ above $\alpha_{dAT}$ we enter different phases depending on the value of $\kappa$:

- For $\kappa_{1RSB} < \kappa < 0$, the system goes into a fRSB phase, which we have described above, through a continuous phase transition.

- For $\kappa_{RFOT} < \kappa < \kappa_{1RSB}$ the system passes into a 1RSB phase, always through a continuous phase transition, before entering at larger value of $\alpha$ into a fRSB phase. In the 1RSB phase, $q(x)$ is a stepwise function, with $q(x) = q_0$ for $x < m$ and $q(x) = q_1$ for $x \geq m$.

- For $\kappa < \kappa_{RFOT}$ the system goes into a sequence of phase transitions that are also encountered in infinite-dimensional theories of glasses and that are known as Random First Order Transitions (RFOT). Firstly, (before RS becomes unstable), for $\alpha_{\mathrm{dyn}} < \alpha < \alpha_K$ the system enters a "*dynamical 1RSB*" phase: although the free energy is equal to that found using an RS Ansatz, the equilibrium measure decomposes into an exponential number of pure states. This corresponds to having an 1RSB phase with $m = 1$. Further increasing $\alpha$ above $\alpha_K$, we cross the *Kauzmann line*, indicating the onset of a 1RSB phase with $m < 1$. Finally, for $\alpha > \alpha_G$ the system enters a *Gardner phase*, where the $q(x)$ exhibits both a

1RSB-like discontinuity at $x = m$, and an fRSB-like continuous part for $x_m \leq x \leq x_M$, with $m \leq x_m$.

## 5.2 Gardner phase, overlap gap and algorithmic implications

It is natural to wonder where the boundary between the fRSB and Gardner phase lies, as this has important algorithmic consequences. Indeed, Refs. [27, 28] analyzed an algorithm called *Incremental AMP* (iAMP) which provably finds a solution in the whole SAT phase, provided that the distribution of overlaps of typical states has a connected support. Throughout the paper we called this the *No Overlap Gap* condition (nOG). This property holds in the fRSB phase, however it does not in the Gardner phase (nor in the 1RSB phase). The boundary between these phases could thus act as an algorithmic threshold, at least for iAMP.

Our contribution is thus to give a numerical estimate of this line, which we call $\alpha_{1+fRSB}$. Rather than looking at the $q(x)$ directly, we use a more precise criterion. Starting in the fRSB phase for a suitable fixed value of $\kappa$, we look at the derivative of $q(x)$, which can be calculated analytically in the region $[x_m, x_M]$ (see appendix D). Then we increase the value of $\alpha$; $\alpha^{1+fRSB}$ corresponds to the first point where $\dot{q}(x_m)$ becomes negative. Solutions with negative derivative are unphysical, so they signal a discontinuity in the function, which corresponds to a gap in the overlap distribution. More details and several plots of $q(x)$ and $\dot{q}(x)$ near the transition to the Gardner phase are reported in Appendix D. Notice that a similar criterion was used in [9] to determine the numerical value of $\kappa_{1RSB}$.

Finally, let us make the following remark about the shape of the $\alpha^{1+fRSB}$ line. For any point under it, iAMP provably finds a solution. However, also every point in the bottom left corner can be solved: indeed if we define the point $(\kappa_*, \alpha_*)$ as the intersection between $\alpha^{1+fRSB}$ and $\alpha_c^{fRSB}$, for any point $(\kappa, \alpha)$ with $\kappa < \kappa_*$ and $\alpha < \alpha_*$, we can simply increase the number of patterns (go vertically in the plot) or increase the margin (go horizontally) such that we are in the fRSB phase, find a solution using iAMP, and this will still be a solution for the original point. In particular, this would suggest that there are parts of the Gardner and 1RSB phase which are solvable, and thus the landscape should not exhibit the Overlap Gap Property. We stress that this is not a contradiction: previous works [20, 26, 40] have shown that it is sometimes possible to find solutions even in phases where the Overlap Gap Condition holds, by looking for atypical solutions.

# 6 Numerical simulations

In this section we compare our estimates of the critical capacity with the performance of Gradient Descent (GD), a first-order optimization method which is a variant of the most widely used optimization algorithm for neural networks, Stochastic Gradient Descent (SGD).

In order to find a solution using GD we used a (differentiable) loss function $\mathcal{L}(\boldsymbol{w})$

$$\mathcal{L}(\boldsymbol{w}) = \sum_{\mu=1}^{\alpha N} \ell(\Delta^\mu(\boldsymbol{w}; \kappa)), \tag{45}$$

where $\ell(\bullet)$ is a loss function per pattern. Generically $\ell(\bullet)$ is chosen to be small if the stability of each pattern in the training set is large and large otherwise. Commonly used loss functions are

$$\ell(x) = \frac{1}{\gamma} \ln(1 + e^{-\gamma x}), \tag{46a}$$

$$\ell(x) = \frac{x^2}{2} \Theta(-x), \tag{46b}$$

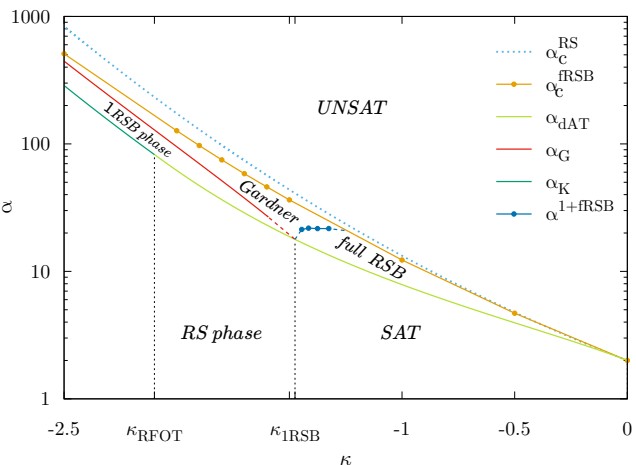

Figure 5: Phase Diagram of the Negative Perceptron. The dynamical transition line $\alpha_{dyn}(\kappa)$ that exists for $\kappa < \kappa_{\text{RFOT}}$ is not displayed for clarity reasons, but it can be found in [20]. Dashed lines represent linear interpolations of the Gardner and 1+fRSB transitions to their intersections with the dAT line which happens at $\kappa = \kappa_{1RSB}$. The dotted line represents the critical capacity evaluated with the RS Ansatz.

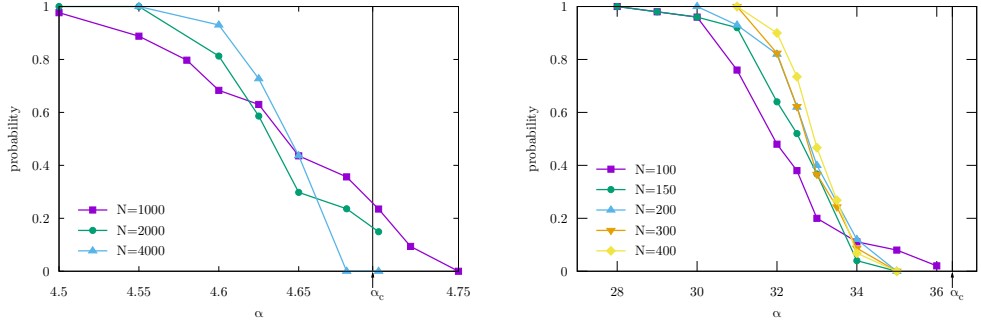

Figure 6: Probability of finding solutions using GD on the cross entropy loss (46a) versus $\alpha$ for the negative perceptron with $\kappa = -0.5$, with sizes $N = 1000, 2000, 4000$ (left panel) and with $\kappa = -1.5$ for $N = 100, 150, 200$ and $300$. In the GD simulations we have fixed the learning rate $\eta = 1$ and the maximum number of training epochs to $2 \cdot 10^6$. The vertical black line represents the exact value of the SAT/UNSAT transition.

that are called respectively the *cross entropy* and *quadratic hinge* loss.

A solution is found by running the following iterative scheme

$$w_{t+1} = w_t - \eta \nabla_w \mathcal{L}(w),$$ (47)

until all constrains $\Delta^\mu(w; \kappa) \geq 0$ for $\mu = 1, \dots, P$ are satisfied. In this model particular attention needs to be paid to the norm, since we are studying the set of solutions subject to the constraint given in equation (3), and the dynamics given by equation (47) will not keep the weights normalized as we want. There are two ways to deal with this:

- Introduce a normalization step after every GD update.

- Keep the norm free to vary, and normalize it when the number of errors is calculated.

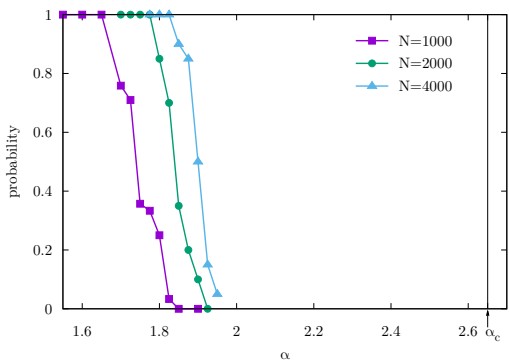

Figure 7: Probability of finding solutions using GD on the cross entropy loss (46a) versus $\alpha$ for the tree-committee machine with a ReLU activation function. Here we have used $K = 100$ and sizes $N = 1000, 2000, 4000$. In the GD simulations we have fixed the learning rate $\eta = 1$ and the maximum number of training epochs to $10^5$. The vertical black line represents the exact value of the SAT/UNSAT transition.

When training a tree-committee machine we have empirically observed that the first method leads to a larger probability of finding a solution, while for the negative perceptron the second method works best.

In Figure 6 we show the probability of finding a solution for a the negative perceptron as a function of $\alpha$ at fixed $\kappa = -0.5, -1.5$ and for several values of $N$. As we can see, as $N$ increases, the transition between non-solvable and solvable problems becomes sharper. This transition, however, clearly happens at values of $\alpha$ below the critical capacity, thus implying that there is an algorithmic gap. Similar conclusions can be drawn in the tree-committee machine case with ReLU activation functions, see Figure 7.

# 7    Conclusions

In the present work we studied the storage problem for two prototypical neural network models, the *Negative Perceptron* and the *Tree-Committee Machine*. Using the replica method, we determined the saddle-point equations that the order parameters need to satisfy, for arbitrary (negative) margin $\kappa$ for the first and for arbitrary activation function $\varphi$ for the latter. Focusing on the *Full-RSB* region of the phase space, we solved these equations numerically using a $k$-RSB Ansatz with large $k$, and used the solutions to compute several observables. By performing a linear fit of the inverse reduced pressure near the SAT/UNSAT threshold we were able to give a high precision numerical estimate of this transition.

For the negative perceptron we determined another novel phase transition between a *fRSB* and *Gardner* phase, and gave a numerical estimate of the value of this threshold. We discussed the *no Overlap Gap condition*, according to which the support of the distribution $P(q)$ of typical states is connected, and identified the boundaries of validity of this property in the phase diagram. The authors of [27] recently proposed an algorithm, *iAMP*, which provably finds solutions under the *nOG* hypothesis. We have showed that this hypothesis does not hold in the *Gardner* phase. This could indicate that this transition acts as an algorithmic threshold for this model.

Finally, we compared our estimates of the SAT-UNSAT threshold with the performance of *Gradient Descent*. In all cases analyzed we have given evidence that Gradient Descent stops finding solutions before the exact SAT/UNSAT threshold that we computed, thus suggesting

the presence of an algorithmic gap. Whether this algorithmic gap is only a barrier to gradient descent or it obstructs also other, smarter, algorithms is an open question that it is left for future work.

## Acknowledgments

E.M.M. and B.L.A. are grateful to Tommaso Rizzo and Luca Leuzzi for many interesting discussions and Riccardo Zecchina for encouragement and advices.

**Funding information** E.M.M. acknowledges the MUR-Prin 2022 funding Prot. 20229T9EAT, financed by the European Union (Next Generation EU).

## A  Properties of $k$-RSB and fRSB matrices

### A.1  Eigenvalues

We derive the eigenvalues of a fRSB matrix by iteration starting from the RS case and moving to the 1 and 2RSB case. For the sake of generality we will suppose the matrix $q^{ab}$ is parameterized by the value it attains on its diagonal $q_d$ and the (step) functions corresponding to values out of the diagonal: $q^{ab} \to \{q_d, q(x)\}$.

- A RS matrix can be decomposed as a sum of two matrices,

$$q^{ab} = (q_d - q)\delta_{ab} + q, \tag{A.1}$$

that commute between each other, so they can be simultaneously diagonalized. An $n \times n$ matrix with all elements equal to $q$ has $n-1$ degenerate zero eigenvalues and one eigenvalue equal to $nq$. We therefore get two eigenvalues

$$\lambda_{-1} = q_d - q + nq, \qquad d_{-1} = 1, \tag{A.2a}$$
$$\lambda_0 = q_d - q, \qquad\qquad d_0 = n - 1. \tag{A.2b}$$

- A 1RSB matrix can be expressed as the sum of 3 terms

$$q^{ab} = (q_d - q_1)\delta_{ab} + (q_1 - q_0)I_{ab}^{m_0} + q_0, \tag{A.3}$$

where $I_{ab}^{m_0}$ is the $n \times n$ matrix having elements equal to 1 inside the blocks of size $m_0$ located around the diagonal and 0 otherwise. Again all the three matrices commute with each other and can be simultaneously diagonalized. Each of the $n/m_0$ blocks of $I_{ab}^{m_0}$ has all equal elements equal to 1, therefore it has $\frac{n}{m_0}(m_0 - 1)$ eigenvalues equal to 0 and $\frac{n}{m_0}$ equal to $m_0$. We therefore have the following eigenvalues

$$\lambda_{-1} = q_d - q_1 + m_0(q_1 - q_0) + nq_0, \qquad d_{-1} = 1, \tag{A.4a}$$
$$\lambda_0 = q_d - q_1 + m_0(q_1 - q_0), \qquad\qquad d_0 = \frac{n}{m_0} - 1 = n\left(\frac{1}{m_0} - \frac{1}{n}\right), \tag{A.4b}$$
$$\lambda_1 = q_d - q_1, \qquad\qquad\qquad\qquad d_1 = \frac{n}{m_0}(m_0 - 1) = n\left(1 - \frac{1}{m_0}\right). \tag{A.4c}$$

- A 2RSB matrix is decomposed as

$$q^{ab} = (q_d - q_2)\delta_{ab} + (q_2 - q_1)I^{m_1}_{ab} + (q_1 - q_0)I^{m_0}_{ab} + q_0, \tag{A.5}$$

repeating the same argument as above we have

$$\lambda_{-1} = q_d - q_2 + m_1(q_2 - q_1) + m_0(q_1 - q_0) + nq_0, \tag{A.6a}$$

$$\lambda_0 = q_d - q_2 + m_1(q_2 - q_1) + m_0(q_1 - q_0), \tag{A.6b}$$

$$\lambda_1 = q_d - q_2 + m_2(q_2 - q_1), \tag{A.6c}$$

$$\lambda_2 = q_d - q_2, \tag{A.6d}$$

with degeneracies respectively

$$d_{-1} = 1, \tag{A.7a}$$

$$d_0 = \frac{n}{m_0} - 1 = n\left(\frac{1}{m_0} - \frac{1}{n}\right), \tag{A.7b}$$

$$d_1 = \frac{n}{m_0}(m_0 - 1) - \frac{n}{m_1}(m_1 - 1) = n\left(\frac{1}{m_1} - \frac{1}{m_0}\right), \tag{A.7c}$$

$$d_2 = n\left(1 - \frac{1}{m_1}\right). \tag{A.7d}$$

- Generalizing to a $k$-RSB matrix we have

$$\lambda_s = \sum_{i=s}^{k} m_i(q_{i+1} - q_i), \qquad d_s = n\left(\frac{1}{m_s} - \frac{1}{m_{s-1}}\right), \qquad s = -1, \dots, k, \tag{A.8}$$

where we have defined $m_k = 1$, $q_{k+1} = q_d$ and $m_{-1} = n \to 0$, $q_{-1} = 0$, $m_{-2} = \infty$. Notice also that in the small $n$ limit $\lambda_{-1} = \lambda_0$.

In the continuous limit the eigenvalues become a function of $x$:

$$\lambda(x) = \int_{q(x)}^{1} dq' x(q') = q_d - xq(x) - \int_x^1 dy\, q(y). \tag{A.9}$$

As in the previous sections, we will denote by $\lambda_m$ and $\lambda_M$ the values of $\lambda$ corresponding to the minimum $q_m$ and a maximum $q_M$ value of the overlap, i.e.

$$\lambda_m = q_d - \int_0^1 dx\, q(x), \tag{A.10a}$$

$$\lambda_M = q_d - q_M. \tag{A.10b}$$

## A.2 Inverse

Since $k$-RSB matrices form a group, the inverse element $p_{ab} = (q^{-1})_{ab}$ must be an element of the group. Therefore the functional form of the eigenvalues is the same as the one derived before. Moreover, we know that the eigenvalues are simply $1/\lambda_s$ with $s = 0, \dots, k$. We therefore have

$$\sum_{i=s}^{k} m_i(p_{i+1} - p_i) = \frac{1}{\sum_{i=s}^{k} m_i(q_{i+1} - q_i)}. \tag{A.11}$$

Those are $k+1$ equations in $k+1$ unknowns. They can be solved iteratively; first of all taking the $i = k$ index we get

$$p_d = p_k + \frac{1}{q_d - q_k}.$$ (A.12)

By subtracting the $(s-1)$-th and the $s$-th equations we get the recursion

$$p_s = p_{s-1} + \frac{1}{m_{s-1}} \left[ \frac{1}{\sum_{i=s-1}^{k} m_i(q_{i+1} - q_i)} - \frac{1}{\sum_{i=s}^{k} m_s(q_{i+1} - q_i)} \right]$$
$$= p_{s-1} + \frac{1}{m_{s-1}} \left( \frac{1}{\lambda_{s-1}} - \frac{1}{\lambda_s} \right) = p_{s-1} - \frac{q_s - q_{s-1}}{\lambda_{s-1}\lambda_s}, \qquad s = 0, \dots, k.$$ (A.13)

Iterating we get that the inverse of a $k$-RSB matrix elements are given by

$$p_s = -\frac{q_0}{\lambda_0^2} - \sum_{i=1}^{s} \frac{q_i - q_{i-1}}{\lambda_{i-1}\lambda_i},$$ (A.14a)

$$p_d = \frac{1}{q_d - q_k} - \frac{q_0}{\lambda_0^2} - \sum_{i=1}^{k} \frac{q_i - q_{i-1}}{\lambda_{i-1}\lambda_i}.$$ (A.14b)

In the $k \to \infty$ limit we therefore get

$$\lim_{k\to\infty} p_s = p(x) = -\frac{q_m}{\lambda_m^2} - \int_0^x dx \frac{\dot{q}(s)}{\lambda^2(s)} = -\frac{q_m}{\lambda_m^2} - \int_{q_m}^{q} \frac{dq'}{\lambda^2(q')},$$ (A.15a)

$$\lim_{k\to\infty} p_d = \frac{1}{q_d - q_M} - \frac{q_m}{\lambda_m^2} - \int_0^1 dx \frac{\dot{q}(s)}{\lambda^2(s)} = \frac{1}{\lambda_M} - \frac{q_m}{\lambda_m^2} - \int_0^1 \frac{dq'}{\lambda^2(q')}.$$ (A.15b)

Notice how the right hand side of the first equation above is equivalent to the left hand side of the saddle point equation (34).

## A.3 Log of the determinant

Having computed the eigenvalues of a generic $k$-RSB matrix with diagonal elements $q_d$, we are now ready to compute the log of the determinant, which appears in the entropic term, see for example (B.3). We are interested as usual in the limit $n \to 0$. We have

$$\lim_{n\to 0} \frac{1}{n} \ln \det q = \lim_{n\to 0} \frac{1}{n} \sum_{i=-1}^{k} d_i \ln \lambda_i$$
$$= \lim_{n\to 0} \left[ \sum_{i=-1}^{k} \frac{1}{m_i} \ln \lambda_i - \sum_{i=-1}^{k-1} \frac{1}{m_i} \ln \lambda_{i+1} \right]$$
$$= \ln(q_d - q_k) + \frac{q_0}{\lambda_0} + \sum_{i=0}^{k-1} \frac{1}{m_i} \ln \frac{\lambda_i}{\lambda_{i+1}}$$ (A.16)
$$= \ln(q_d - q_k) + \frac{q_0}{\lambda_0} + \sum_{i=0}^{k-1} \frac{1}{m_i} \ln \left( 1 + \frac{m_i(q_{i+1} - q_i)}{\lambda_{i+1}} \right).$$

When $k$ is large $q_i - q_{i-1}$ is small, so that, in the continuous limit, we get

$$\lim_{k\to\infty} \lim_{n\to 0} \frac{1}{n} \ln \det q = \ln(q_d - q_M) + \frac{q_m}{\lambda_m} + \int_{x_m}^{x_M} dx \frac{\dot{q}(x)}{\lambda(x)}.$$ (A.17)

## A.4 Asymptotic behaviour of $f(m_l, h)$

We start from the recursion relation in the case of the number of error loss function

$$f(m_k, h) = f(x_M, h) = \ln \int dz\, \mathcal{N}_{\Delta(1)-\Delta(q_k)}(z+h)\, e^{-\beta \ell(z-\kappa)} = \ln H\left(\frac{\kappa + h}{\sqrt{\Delta(1)-\Delta(q_k)}}\right),$$

$$f(m_s, h) = \frac{1}{m_s} \ln \int dz\, \mathcal{N}_{\Delta(q_{s+1})-\Delta(q_s)}(z-h)\, e^{m_s f(m_{s+1}, z)}, \qquad s = k-1, \ldots, 0. \tag{A.18}$$

We know that

$$\ln H(x) \simeq -\frac{x^2}{2} - \ln x - \frac{1}{2}\ln(2\pi), \qquad \text{as } x \to +\infty, \tag{A.19}$$

whereas it goes exponentially to 0 as $x \to -\infty$. Therefore

$$\ln H\left(\frac{\kappa + h}{\sqrt{\Delta(1)-\Delta(q_k)}}\right) = \begin{cases} -\frac{(\kappa+h)^2}{2(\Delta(1)-\Delta(q_k))} \equiv -\frac{(\kappa+h)^2}{2\Lambda_k}, & h \to +\infty, \\ O(e^{-h^2}), & h \to -\infty, \end{cases} \tag{A.20}$$

where $\Lambda_k \equiv \Delta(1) - \Delta(q_k)$. Similarly we will define the quantities

$$\Lambda_s = \sum_{i=s}^{k} m_i (\Delta(q_{i+1}) - \Delta(q_i)) = \Lambda_{s+1} + m_s(\Delta(q_{s+1}) - \Delta(q_s)), \qquad s = -1, \ldots, k, \tag{A.21}$$

which will appear naturally in the following, and which represent the eigenvalues of the effective order parameter matrix $\Delta_{ab}$.

The asymptotic behavior of $f(m_k, h)$ at at $h \to \pm\infty$ will induce a similar one for the functions $f(m_s, h)$ with $s = k-1, \ldots, 0$. Let's start with the case $s = k-1$. We have for $h \to \infty$

$$\begin{aligned}
f(m_{k-1}, h) &= \frac{1}{m_{k-1}} \ln \int dz\, \mathcal{N}_{\Delta(q_k)-\Delta(q_{k-1})}(z)\, e^{m_{k-1} f(m_k, z+h)} \\
&= \frac{1}{m_{k-1}} \ln \int_{-h}^{\infty} dz\, \mathcal{N}_{\Delta(q_k)-\Delta(q_{k-1})}(z)\, e^{-\frac{m_{k-1}(\kappa+z+h)^2}{2(\Delta(1)-\Delta(q_k))}} \\
&\simeq \frac{1}{m_{k-1}} \ln \int_{-h}^{\infty} dz\, e^{-\frac{z^2}{2(\Delta(q_k)-\Delta(q_{k-1}))} - \frac{m_{k-1}(\kappa+z+h)^2}{2(\Delta(1)-\Delta(q_k))}} \\
&\simeq -\frac{(\kappa+h)^2}{2(\Delta(1)-\Delta(q_k))} + \frac{1}{m_{k-1}} \ln \int_{-h}^{\infty} dz\, e^{-a_k z^2 - b_k z},
\end{aligned} \tag{A.22}$$

where we have neglected low order terms in $h$ and defined the quantities

$$a_k \equiv \frac{1}{2}\left(\frac{m_{k-1}}{\Delta(1)-\Delta(q_k)} + \frac{1}{\Delta(q_k)-\Delta(q_{k-1})}\right) = \frac{m_{k-1}\Lambda_{k-1}}{2\Lambda_k(\Lambda_{k-1}-\Lambda_k)}, \tag{A.23a}$$

$$b_k \equiv \frac{m_{k-1}(\kappa+h)}{\Delta(1)-\Delta(q_k)} = \frac{m_{k-1}(\kappa+h)}{\Lambda_k}. \tag{A.23b}$$

Using the identity

$$\int_{\gamma}^{+\infty} dz\, e^{-\alpha z^2 - \beta z} = \sqrt{\frac{\pi}{\alpha}}\, e^{\frac{\beta^2}{4\alpha}} H\left(\frac{\beta + 2\alpha\gamma}{\sqrt{2\alpha}}\right), \tag{A.24}$$

and noticing that the argument of the $H$ function $b_k - 2a_k h = -\frac{h}{q_k - q_{k-1}} \to -\infty$ we have

$$
\begin{aligned}
f(m_k, h) &\simeq -\frac{(\kappa + h)^2}{2(\Delta(1) - \Delta(q_k))} + \frac{1}{m_{k-1}} \ln\left[ e^{\frac{b_k^2}{4a_k}} H\left( \frac{b_k - 2a_k h}{\sqrt{2a_k}} \right) \right] \\
&= -\frac{(\kappa + h)^2}{2(\Delta(1) - \Delta(q_k))} + \frac{b_k^2}{4m_{k-1}a_k} \\
&= -\frac{(\kappa + h)^2}{2\Lambda_k} + \frac{m_{k-1}(\kappa + h)^2(\Lambda_{k-1} - \Lambda_k)}{2\Lambda_k \Lambda_{k-1}} \\
&= -\frac{(\kappa + h)^2}{2\left(\Delta(1) - \Delta(q_k) + m_{k-1}(\Delta(q_k) - \Delta(q_{k-1}))\right)} \\
&\equiv -\frac{(\kappa + h)^2}{2\Lambda_k}.
\end{aligned}
\tag{A.25}
$$

Iterating we get, for $s = 0, \ldots, k$

$$
f(m_s, h) \simeq -\frac{(\kappa + h)^2}{2\Lambda_s}, \qquad \text{as } h \to +\infty. \tag{A.26}
$$

Notice that since $\Delta(q_{s+1}) \geq \Delta(q_s)$, then $\Lambda_s \geq \Lambda_{s+1}$ for all $s = 0, \ldots, k$; this tells us that $f(m_s, h)$ diverges slower to $-\infty$ for $h \to +\infty$ with respect to $f(m_{s+1}, h)$. Similarly one finds that

$$
f(m_s, h) \simeq O(e^{-h^2}), \qquad \text{as } h \to -\infty. \tag{A.27}
$$

# B $k$-steps replica symmetry breaking Ansatz

In this first appendix we derive the expressions of the entropic and energetic term for finite number of breakings of Replica Symmetry [44, 45, 54], and we mention how we have solved the corresponding saddle point equations. We remind that we call the $k + 1$ values assumed by the matrix $q^{ab}$ as $q_0, q_1, \ldots, q_k$, and the block sizes respectively as $m_0, m_1, \ldots, m_{k-1}$. We will use the square bracket notation $[\bullet]_s$ to denote the operation of extracting step $s + 1$ from the $k$-step RSB matrix in its argument, i.e., for example, $[q^{ab}]_s = q_s$.

## B.1 Entropic potential

Imposing the $k$-RSB structure on the overlap matrix $q^{ab}$ will enable us to perform the small $n$ limit

$$
\phi = \max_{\boldsymbol{q}} \mathcal{S}(\boldsymbol{q}), \tag{B.1a}
$$

$$
\mathcal{S}(\boldsymbol{q}) = \mathcal{G}_S(\boldsymbol{q}) + \alpha \mathcal{G}_E(\boldsymbol{q}) \equiv \lim_{n \to 0} \frac{G_S(\boldsymbol{q})}{n} + \alpha \lim_{n \to 0} \frac{G_E(\boldsymbol{q})}{n}. \tag{B.1b}
$$

In order to compute the entropic term $\mathcal{G}_S(\boldsymbol{q})$ one needs to compute the eigenvalues of a generic $k$-RSB matrix and the corresponding multiplicities. In appendix A.1 we show that there are $k + 2$ eigenvalues $\lambda_s$ with multiplicities $d_s$, $s = -1, 0, \ldots, k$ which read

$$
\lambda_s = \sum_{i=s}^{k} m_i(q_{i+1} - q_i), \qquad d_s = n\left( \frac{1}{m_s} - \frac{1}{m_{s-1}} \right), \qquad s = -1, \ldots, k. \tag{B.2}
$$

In the previous equations we have used the definitions $m_k = 1$, $q_{k+1} = 1$ and $m_{-1} = n$, $q_{-1} = 0$, $m_{-2} = \infty$. Once the eigenvalues are known, one can compute the entropic term,

which consists in computing the log of the determinant of $\boldsymbol{q}$ in the small $n$ limit. We show in appendix A.3 that it reads

$$\mathcal{G}_S(\boldsymbol{q}) = \lim_{n\to 0} \frac{1}{2n} \ln \det \boldsymbol{q} = \frac{1}{2} \ln(1 - q_k) + \frac{q_0}{2\lambda_0} + \sum_{i=0}^{k-1} \frac{1}{2m_i} \ln\left(1 + \frac{m_i(q_{i+1} - q_i)}{\lambda_{i+1}}\right). \tag{B.3}$$

### B.2 Infinite width energetic potential

#### B.2.1 Derivation of equation (18)

In this first section we are going to derive the energetic term in the large $K$ limit, deriving explicitly equation (18). We report here the starting energetic term expression (14c)

$$G_E(\boldsymbol{q}) \equiv \ln \mathbb{E}_y \int \prod_{la} \frac{d\lambda_l^a d\hat{\lambda}_l^a}{2\pi} \prod_a e^{-\beta\ell\left(\frac{y}{\sqrt{K}} \sum_{l=1}^K c_l \varphi(\lambda_l^a) - \kappa\right)} e^{i\sum_{la} \lambda_l^a \hat{\lambda}_l^a - \frac{1}{2}\sum_{ab,l} q_l^{ab} \hat{\lambda}_l^a \hat{\lambda}_l^b}. \tag{B.4}$$

The first step is to extract the variables $u_a \equiv \frac{1}{\sqrt{K}} \sum_{l=1}^K c_l \varphi(\lambda_l^a)$ from the loss function via $n$ delta functions and inserting their integral representations

$$
\begin{aligned}
G_E(\boldsymbol{q}) &= \ln \mathbb{E}_y \int \prod_a \frac{du_a d\hat{u}_a}{2\pi} e^{i\sum_a u_a \hat{u}_a} \int \prod_{la} \frac{d\lambda_l^a d\hat{\lambda}_l^a}{2\pi} \prod_a e^{-\beta\ell(yu_a - \kappa)} \\
&\quad \times e^{i\sum_{la} \lambda_l^a \hat{\lambda}_l^a - \frac{1}{2}\sum_{ab,l} q_l^{ab} \hat{\lambda}_l^a \hat{\lambda}_l^b - i\sum_a \hat{u}_a \frac{1}{\sqrt{K}} \sum_{l=1}^K c_l \varphi(\lambda_l^a)} \\
&= \ln \mathbb{E}_y \int \prod_a \frac{du_a d\hat{u}_a}{2\pi} e^{i\sum_a u_a \hat{u}_a - \beta\sum_a \ell(yu_a - \kappa)} \prod_{l=1}^K \left\langle e^{-i\sum_a \hat{u}_a \frac{c_l}{\sqrt{K}} \varphi(\lambda_l^a)} \right\rangle,
\end{aligned}
\tag{B.5}
$$

where we have introduced the notation

$$\langle \bullet \rangle \equiv \int \prod_a \frac{d\lambda_l^a d\hat{\lambda}_l^a}{2\pi} e^{i\sum_a \lambda_l^a \hat{\lambda}_l^a - \frac{1}{2}\sum_{ab} q_l^{ab} \hat{\lambda}_l^a \hat{\lambda}_l^b} \bullet . \tag{B.6}$$

The next step is to Taylor expand in $1/\sqrt{K}$ the term $\prod_{l=1}^K \left\langle e^{-i\sum_a \hat{u}_a \frac{c_l}{\sqrt{K}} \varphi(\lambda_l^a)} \right\rangle$, averaging term by term and then re-exponentiate. It suffices to maintain the terms up to second order in the expansion, since the other are negligible in the large $K$ limit

$$
\begin{aligned}
\prod_{l=1}^K \left\langle e^{-i\sum_a \hat{u}_a \frac{c_l}{\sqrt{K}} \varphi(\lambda_l^a)} \right\rangle &= \prod_{l=1}^K \left[ 1 - \frac{ic_l}{\sqrt{K}} \sum_a \hat{u}_a \left\langle \varphi(\lambda_l^a) \right\rangle - \frac{c_l^2}{2K} \sum_{ab} \hat{u}_a \hat{u}_b \left\langle \varphi(\lambda_l^a) \varphi(\lambda_l^b) \right\rangle + O\left(\frac{1}{K^{3/2}}\right) \right] \\
&= e^{-i\sum_a \hat{u}_a M_a - \frac{1}{2}\sum_{ab} \Delta_{ab} \hat{u}_a \hat{u}_b},
\end{aligned}
\tag{B.7}
$$

where $M_a$ and $\Delta_{ab}$ are the quantities defined in (19a). Notice that this equivalent to say that the variable $u_a = \frac{1}{\sqrt{K}} \sum_{l=1}^K c_l \varphi(\lambda_l^a)$ is distributed as a multivariate Gaussian with mean $M_a$ and covariance matrix $\Delta_{ab}$: indeed the left hand side of the previous equation is exactly the cumulant generating function of the random variable $u_a$. Inserting this identity back in equation (B.5) one gets (18).

#### B.2.2 Effective order parameters

If we impose a $k$-RSB Ansatz on $q^{ab}$, also the effective order parameters $\Delta_{ab}$,

$$\Delta_{ab} = e^{\frac{1}{2}\sum_{ab} q^{ab} \frac{\partial^2}{\partial s_a \partial s_b}} \varphi(s_a) \varphi(s_b) \Big|_{s_a = 0}, \tag{B.8}$$

will be a $k$-steps RSB matrix with the same block size as $q^{ab}$. It is easy to show that the $s+1$ step of $\Delta_{ab}$ is

$$
\begin{aligned}
\left[\Delta^{ab}\right]_s &= \int Dz_0 \ldots Dz_s \left[\int Dz_{s+1} \ldots Dz_{k+1} \, \varphi\left(\sum_{l=0}^{k+1} \sqrt{q_l - q_{l-1}} z_l\right)\right]^2 \\
&= \int Dx \left[\int Dy \, \varphi\left(\sqrt{q_s}\, x + \sqrt{1-q_s}\, y\right)\right]^2 \equiv \Delta(q_s),
\end{aligned}
\tag{B.9}
$$

i.e. it depends on $q_s$ only. We have used in the second line the fact that the sum of Gaussian random variables is still Gaussian distributed (or equivalently, we have performed several 2-dimensional rotations over the variables $z_0 \ldots z_s$ and $z_{s+1}, \ldots, z_{k+1}$). Notice that the previous expression can also be written as a two dimensional Gaussian integral

$$
\Delta(q_s) = \int \frac{d\boldsymbol{h}}{2\pi\sqrt{\det\mathcal{C}}} \, e^{-\frac{1}{2}\boldsymbol{h}^T \mathcal{C}^{-1} \boldsymbol{h}} \, \varphi(h_1)\varphi(h_2),
\tag{B.10}
$$

where

$$
\mathcal{C} = \begin{pmatrix} 1 & q_s \\ q_s & 1 \end{pmatrix},
\tag{B.11}
$$

therefore showing that our effective order parameters are also equivalent to the NNGP kernel appearing in neural networks at initialization or in the lazy regime. In the following we will also need the indices $l = -1$ and $l = k+1$ in order to write down the expression of the entropy; consistently with the notation $q_{-1} \equiv 0$ and $q_{k+1} \equiv 1$, they can be found by substituting them in (B.9), i.e.

$$
\Delta(q_{-1}) = \Delta(0) = \left[\int Dx \, \varphi(x)\right]^2,
\tag{B.12a}
$$

$$
\Delta(q_{k+1}) = \Delta(1) = \int Dx \, \varphi^2(x).
\tag{B.12b}
$$

Given those definitions the energetic term reads

$$
\mathcal{G}_E = \frac{1}{m_0}\int Dz_0 \ln \int Dz_1 \left[\ldots \left[\int Dz_{k+1} e^{-\beta\ell\left(\sqrt{\Delta(1)-\Delta(q_k)} z_{k+1} - \sum_{s=0}^{k}\sqrt{\Delta(q_s)-\Delta(q_{s-1})} z_s - \kappa\right)}\right]^{\frac{m_{k-1}}{m_k}} \ldots\right]^{\frac{m_0}{m_1}}.
\tag{B.13}
$$

The energetic term can be written more compactly defining a discrete set of functions $f(m_s, h)$, with $s = 0, \ldots, k$, that satisfy the iterative rule

$$
\begin{aligned}
f(m_k, h) &= \ln \int dz \, \mathcal{N}_{\Delta(1)-\Delta(q_k)}(z+h) \, e^{-\beta\ell(z-\kappa)}, \\
f(m_s, h) &= \frac{1}{m_s}\ln \int dz \, \mathcal{N}_{\Delta(q_{s+1})-\Delta(q_s)}(z-h) e^{m_s f(m_{s+1},z)}, \qquad s = k-1, \ldots, 0,
\end{aligned}
\tag{B.14}
$$

where $\mathcal{N}_\sigma(z) \equiv \frac{e^{-\frac{z^2}{2\sigma}}}{\sqrt{2\pi\sigma}}$. Notice how the iteration rule for $\tilde{f}(m_0, h) \equiv f(m_0, -h-\kappa)$ does not explicitly depend on $\kappa$ (this the function that is actually used in [9]). Notice that in error counting loss, which we focus on in this paper, $\ell(x) = \Theta(-x)$ the integral in the initial condition for $f$ can be explicitly solved, giving

$$
f(m_k, h) = \ln H_\beta\left(\frac{\kappa + h}{\sqrt{\Delta(1)-\Delta(q_k)}}\right),
\tag{B.15}
$$

where $H_\beta(x) \equiv e^{-\beta} + (1 - e^{-\beta})H(x)$ and $H(x) \equiv \int_x^\infty Dy = \frac{1}{2}\text{Erfc}\left(\frac{x}{\sqrt{2}}\right)$. The energetic term therefore can be expressed in terms of $f(m_0, h)$ as

$$\mathcal{G}_E = \int dh \, \mathcal{N}_{\Delta(q_0) - \Delta(0)}(h) f(m_0, h). \tag{B.16}$$

### B.2.3 Effective order parameters for some activation functions

We list here the expressions of the effective order parameters for some activation functions of interest

- $\varphi(x) = x$, in the case of the identity activation we get back the perceptron case

$$\Delta(q) = q. \tag{B.17}$$

- $\varphi(x) = \text{sign}(x)$ [12, 13, 29]

$$\Delta(q) = 1 - \frac{2}{\pi}\arccos(q). \tag{B.18}$$

- $\varphi(x) = \text{ReLU}(x) = \max(0, x)$

$$\Delta(q) = \frac{\sqrt{1 - q^2}}{2\pi} + \frac{q}{\pi}\arctan\left(\sqrt{\frac{1+q}{1-q}}\right). \tag{B.19}$$

- $\varphi(x) = \text{Erf}(\gamma x)$

$$\Delta(q) = 1 - \frac{2}{\pi}\arccos\left(\frac{2\gamma^2 q}{1 + 2\gamma^2}\right). \tag{B.20}$$

### B.2.4 Alternative approach

One can find (B.13) directly imposing the $k$-RSB Ansatz on finite width version of the energetic term, which reads

$$\mathcal{G}_E = \frac{1}{m_0}\mathbb{E}_y \int \prod_l Dz_l^0 \ln \int \prod_l Dz_l^1 \left[\cdots\left[\int \prod_l Dz_l^{k+1} e^{-\beta\ell\left(\frac{y}{\sqrt{K}}\sum_{l=1}^K c_l \varphi\left(\sum_{s=0}^{k+1}\sqrt{q_s - q_{s-1}}z_l^s\right) - \kappa\right)}\right]^{\frac{m_{k-1}}{m_k}}\cdots\right]^{\frac{m_0}{m_1}}. \tag{B.21}$$

We can now use the central limit theorem repeatedly on this expression to perform the large $K$ limit. We specialize here for simplicity to the number of error loss with $\beta \to \infty$, but the argument can be trivially generalized to generic loss functions. The innermost $K$-dimensional integrals can be simplified as

$$\int \prod_l Dz_l^{k+1} \Theta\left(\frac{y}{\sqrt{K}}\sum_{l=1}^K c_l \varphi\left(\sum_{s=0}^{k+1}\sqrt{q_s - q_{s-1}}z_l^s\right) - \kappa\right) \simeq \int Dz^{k+1}\Theta\left(yM^{(0)} + \sqrt{\Delta^{(0)}}z^{k+1} - \kappa\right)$$

$$= H\left(\frac{\kappa + yM^{(0)}}{\sqrt{\Delta^{(0)}}}\right), \tag{B.22}$$

where $M^{(0)}$ and $\Delta^{(0)}$ are respectively the mean and the variance with respect to variables $z^{k+1}$

$$M^{(0)} \equiv \frac{1}{\sqrt{K}} \sum_{l=1}^{K} c_l \int Dh \, \varphi \left( \sum_{s=0}^{k} \sqrt{q_s - q_{s-1}} z_l^s + \sqrt{1-q_k} h \right), \tag{B.23a}$$

$$\Delta^{(0)} \equiv \frac{1}{K} \sum_{l=1}^{K} c_l^2 \left[ \int Dh \, \varphi^2 \left( \sum_{s=0}^{k} \sqrt{q_s - q_{s-1}} z_l^s + \sqrt{1-q_k} h \right) \right.$$
$$\left. - \left( \int Dh \, \varphi \left( \sum_{s=0}^{k} \sqrt{q_s - q_{s-1}} z_l^s + \sqrt{1-q_k} h \right) \right)^2 \right]. \tag{B.23b}$$

Iterating the procedure $k$ times we have

$$\mathcal{G}_E = \frac{1}{m_0} \mathbb{E}_y \int Dz_0 \ln \int Dz_1 \left[ \ldots \left[ \int Dz_k H^{m_{k-1}} \left( \frac{\kappa + yM + \sum_{s=0}^{k} \sqrt{\Delta(q_s) - \Delta(q_{s-1})} z_s}{\sqrt{\Delta(1) - \Delta(q_k)}} \right) \right]^{\frac{m_{k-2}}{m_{k-1}}} \ldots \right]^{\frac{m_0}{m_1}}, \tag{B.24}$$

where $\Delta(q)$ is the same kernel function defined in (B.9) and the mean term is

$$M \equiv m_c \int Dx \, \varphi(x), \tag{B.25}$$

where $m_c = \frac{1}{\sqrt{K}} \sum_l c_l$.

## B.3 Saddle point equations

The aim of this section is to write the saddle point equations,

$$q_{cd}^{-1} = -\alpha \frac{\partial \Delta_{cd}}{\partial q_{cd}} \frac{e^{\frac{1}{2}\sum_{ab}\Delta_{ab}\frac{\partial^2}{\partial h_a \partial h_b}} \frac{\partial^2}{\partial h_c \partial h_d} \prod_a e^{-\beta \ell(h_a - \kappa)} \Big|_{h_a=0}}{e^{\frac{1}{2}\sum_{ab}\Delta_{ab}\frac{\partial^2}{\partial h_a \partial h_b}} \prod_a e^{-\beta \ell(h_a - \kappa)} \Big|_{h_a=0}} \equiv -\alpha \frac{\partial \Delta_{cd}}{\partial q_{cd}} M_{cd}, \tag{B.26a}$$

$$\frac{\partial \Delta_{cd}}{\partial q_{cd}} = e^{\frac{1}{2}\sum_{ab}q^{ab}\frac{\partial^2}{\partial s_a \partial s_b}} \frac{\partial \varphi(s_c)}{\partial s_c} \frac{\partial \varphi(s_d)}{\partial s_d} \Big|_{s_a=0}, \tag{B.26b}$$

in the $k$-RSB Ansatz in a compact form suitable for numerical evaluations. In the $k$-RSB Ansatz, $\frac{\partial \Delta_{cd}}{\partial q_{cd}}$, $(q^{-1})_{cd}$ and $M_{cd}$ will be $k$-RSB matrices as well. Therefore, in order to compute the update for the overlap $q_s$, we need to compute the matrix elements $\left[q^{-1}\right]_s$, $[M]_s = M_s$ and $\left[\frac{\partial \Delta_{cd}}{\partial q_{cd}}\right]_s$ with $s = 0, \ldots, k$. We start from $\left[\frac{\partial \Delta_{cd}}{\partial q_{cd}}\right]_s$ which is

$$\left[\frac{\partial \Delta_{cd}}{\partial q_{cd}}\right]_s = \int Dx \left[ \int Dy \, \varphi'\left(\sqrt{q_s} x + \sqrt{1-q_s} y\right) \right]^2 = \dot{\Delta}(q_s), \qquad s = 0, \ldots, k, \tag{B.27}$$

having denoted by a dot the derivative with respect to $q$. The matrix elements of $M_{cd}$ instead can be written as

$$M_s = \int dh \, P(m_s, h) f'(m_s, h)^2, \qquad s = 0, \ldots, k, \tag{B.28}$$

where we have denoted with a prime a derivative with respect to $h$. $P$ is instead found by the following iteration rule

$$P(m_{-1}, h) = \delta(h),$$

$$P(m_0, h) = e^{m_{-1}f(m_0,h)} \int dz \, \mathcal{N}_{\Delta(q_0)-\Delta(q_{-1})}(z-h) P(m_{-1}, z) e^{-m_{-1}f(m_{-1},z)} = \mathcal{N}_{\Delta(q_0)-\Delta(0)}(h),$$

$$P(m_l, h) = e^{m_{l-1}f(m_l,h)} \int dz \, \mathcal{N}_{\Delta_{q_l}-\Delta_{q_{l-1}}}(z-h) P(m_{l-1}, z) e^{-m_{l-1}f(m_{l-1},z)}, \qquad l = 1, \dots, k,$$

(B.29)

which is the same as Sherrington Kirkpatrick (SK) model, apart for the effective order parameters.

Finally we can get the update for the steps $q_s$, $s = 0, \dots, k$ by computing the inverse elements of the computed matrix $p_s \equiv -\alpha \dot{\Delta}(q_s) M_s$, $s = 0, \dots, k$. The inverse elements of a generic $k$-RSB matrix with diagonal elements $p_{k+1} \equiv p_d$ are reported in section A.2.

However in order to use those results, we need to know what is the diagonal value assumed by the $k$-RSB matrix $\boldsymbol{p}$, i.e. $p_{k+1} = p_d$. This can be computed knowing that the corresponding diagonal value of the overlap matrix $\boldsymbol{q}$ is $q_{k+1} = q_d = 1$. Therefore we can find $p_d$ by exploiting equation (A.14); in the end one has to solve the implicit equation

$$1 = \frac{1}{p_d - p_k} - \frac{p_0}{\hat{\lambda}_0^2} - \sum_{s=0}^{k-1} \frac{p_{s+1} - p_s}{\hat{\lambda}_s \hat{\lambda}_{s+1}},$$

(B.30)

where $\hat{\lambda}_s$ are the eigenvalues of the matrix $p$

$$\hat{\lambda}_s \equiv \sum_{i=s}^{k} m_i (p_{i+1} - p_i), \qquad s = 0, \dots, k.$$

(B.31)

Once $p_d$ is computed we can find the corresponding values of $q_s$, using the recursions

$$q_s = q_{s-1} - \frac{p_s - p_{s-1}}{\hat{\lambda}_{s-1} \hat{\lambda}_s}, \qquad s = 0, \dots, k,$$

(B.32)

as derived in section A.2.

### B.3.1 Summary

To summarize, in order to solve the $k$-RSB saddle point equations, we use the following procedure. We start with an initial guess for $q_s$, $s = 0, \dots, k$ and a starting value for the minimal value and maximal value of $x$, $x_m = m_0$ $x_M = m_{k-1}$. We generate a grid of $k-2$ points between $x_m$ and $x_M$, given by $m_1 < \dots < m_{k-2}$; the grid need not to be necessary equispaced. Then

1. Compute the effective order parameters $\Delta(q_s)$ and their derivatives $\dot{\Delta}(q_s)$ for $s = 0, \dots, k$ using respectively (B.9), (B.27).

2. Compute $f(m_s, h)$ for $s = k, \dots, 0$ using (B.14) and $P(m_s, h)$ for $s = 0, \dots, k$ using equations (B.29).

3. Compute $M_s$ using (B.28) and then $p_s = -\alpha \dot{\Delta}(q_s) M_s$ with $s = 0, \dots, k$.

4. Compute $p_d$ by solving the implicit equation (B.30).

5. Use relations (B.32) to get a new estimate of $q_s$ from $p_s$.

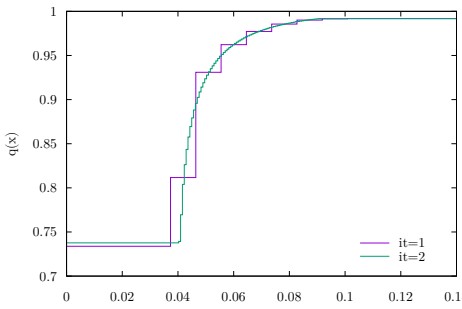
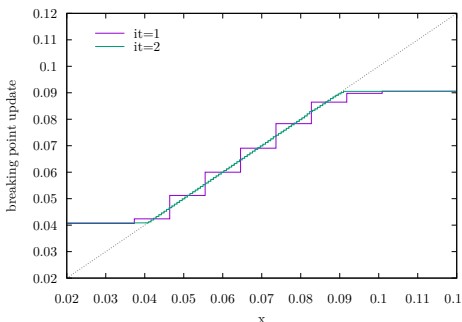

Figure 8: Left panel: $q(x)$ as a function of $x$ before the first and second update of the breaking points (respectively violet and green line) as described in the text. Right panel: breaking point update (i.e. the right hand side of equation (40)) as a function of $x$. Here we have used $\varphi(h) = \text{erf}(h)$ with $\kappa = 0$ and $\alpha = 2.3$. We initialized the code with $x_m = 0.001$ and $x_M = 0.9$ and we used $k = 100$. Only two iterations are sufficient to get a very precise estimate of $x_m$ and $x_M$, i.e. the points where green line departs from the identity (dashed).

6. Repeat points 1-5 until convergence.

7. Update the value of the minimal and maximal breaking-point evaluating (40) respectively in $m_0$ and $m_{k-1}$. Generate a new grid of values of $k - 2$ points between $x_m$ and $x_M$, given by $m_1 < \cdots < m_{k-2}$ and compute the values of $q_s$, $s = 0, \ldots, k$, interpolating the $q(x)$ obtained at point 6.

8. Repeat points 1-7 until convergence.

Once convergence is reached, we can compute all the observable of interest, in particular the free entropy as

$$\phi = \frac{1}{2}\ln(1-q_k) + \frac{q_0}{2\lambda_0} + \sum_{s=0}^{k-1}\frac{1}{2m_s}\ln\left(1 + \frac{m_s(q_{s+1}-q_s)}{\lambda_{s+1}}\right) + \alpha\int dh\, \mathcal{N}_{\Delta(q_0)-\Delta(0)}(h)f(m_0,h). \quad \text{(B.33)}$$

In the left panel of Fig. 8, in the case $\varphi(h) = \text{erf}(h)$, $\kappa = 0$ and $\alpha = 2.3$, we show the plot of $q(x)$ before and after updating the breaking points for the first time. On the right panel we show the corresponding update of the breaking points. It is evident that the convergence on the breaking point is reached very rapidly and most of the situations only two repetitions of points 1-5 are needed.

## B.4 Replica symmetric Ansatz

### B.4.1 Entropy and saddle point equations

In the Replica Symmetric (RS) approximation we have, in the infinite $\beta$ limit the following entropy

$$\phi = \mathcal{G}_S + \alpha\mathcal{G}_E, \quad \text{(B.34a)}$$

$$\mathcal{G}_S = \frac{1}{2(1-q)} + \frac{1}{2}\ln(1-q), \quad \text{(B.34b)}$$

$$\mathcal{G}_E = \int Dz_0 \ln H\left(\frac{\kappa + \sqrt{\Delta(q)-\Delta(0)}z_0}{\sqrt{\Delta(1)-\Delta(q)}}\right). \quad \text{(B.34c)}$$

The corresponding saddle point equation for the overlap $q$ reads

$$\frac{q}{2(1-q)^2} = -\alpha \frac{\partial \mathcal{G}_E}{\partial q} = \alpha \dot{\Delta}(q) \int Dz_0 \left[ \frac{d}{dz} \ln H\left( \frac{z}{\sqrt{\Delta(1)-\Delta(q)}} \right) \Bigg|_{z=\kappa+\sqrt{\Delta(q)-\Delta(0)}z_0} \right]^2$$
$$= \frac{\alpha \dot{\Delta}(q)}{\Delta(1)-\Delta(q)} \int Dz_0\, GH^2\left( \frac{\kappa + \sqrt{\Delta(q)-\Delta(0)}z_0}{\sqrt{\Delta(1)-\Delta(q)}} \right). \tag{B.35}$$

### B.4.2 dAT instability

Applying the RS Ansatz on (36) will allow us to derive the instability of the RS Ansatz itself, know as dAT instability. In this case $\lambda(q) = 1-q$ and the solution to the PDEs in equations (31b) and (35b) is trivial

$$P(q,h) = N_{\Delta(q)-\Delta(0)}(h), \tag{B.36a}$$

$$f(q,h) = \ln H\left( \frac{\kappa+h}{\sqrt{\Delta(1)-\Delta(q)}} \right). \tag{B.36b}$$

Inserting those identities in (39) we get

$$\frac{1}{(1-q)^2} = \alpha \ddot{\Delta}(q) \int Dh \left[ \frac{d}{dz} \ln H\left( \frac{z}{\sqrt{\Delta(1)-\Delta(q)}} \right) \Bigg|_{z=\kappa+\sqrt{\Delta(q)-\Delta(0)}h} \right]^2$$
$$+ \alpha \dot{\Delta}^2(q) \int Dh \left[ \frac{d^2}{dz^2} \ln H\left( \frac{z}{\sqrt{\Delta(1)-\Delta(q))}} \right) \Bigg|_{z=\kappa+\sqrt{\Delta(q)-\Delta(0)}h} \right]^2 \tag{B.37}$$
$$= \frac{\alpha \ddot{\Delta}(q)}{\Delta(1)-\Delta(q)} \int Dh\, GH^2\left( \frac{\kappa + \sqrt{\Delta(q)-\Delta(0)}h}{\sqrt{\Delta(1)-\Delta(q)}} \right)$$
$$+ \frac{\alpha \dot{\Delta}^2(q)}{(\Delta(1)-\Delta(q))^2} \int Dh\, \mathcal{W}^2\left( \frac{\kappa + \sqrt{\Delta(q)-\Delta(0)}h}{\sqrt{\Delta(1)-\Delta(q)}} \right),$$

where

$$\mathcal{W}(z) \equiv \frac{d^2}{dz^2} \ln H(z) = -\frac{d}{dz} GH(z) = GH(z)(z - GH(z)). \tag{B.38}$$

### B.4.3 SAT/UNSAT transition in the RS approximation

To find the SAT/UNSAT transition in the RS approximation we have to perform the $q \to 1$ limit. As evinced in [25], in most of the activation functions, the kernel $\Delta(q)$ scales as

$$\Delta(q) \simeq \Delta(1) - \dot{\Delta}(1)\delta q, \tag{B.39}$$

with $\delta q = 1-q$.

Using the fact that $\ln H(x) \simeq -\frac{1}{2} \ln(2\pi) - \ln x - \frac{x^2}{2}$ as $x \to \infty$, retaining only the most divergent terms we get

$$\int Dz_0 \ln H\left( \frac{\kappa + \sqrt{\Delta(q)-\Delta(0)}z_0}{\sqrt{\Delta(1)-\Delta(q)}} \right) \simeq \int_{-\frac{\kappa}{\sqrt{\Delta(1)-\Delta(0)}}}^{+\infty} Dz_0 \left[ \frac{1}{2} \ln \delta q - \frac{\left(\kappa + \sqrt{\Delta(1)-\Delta(0)}z_0\right)^2}{2\dot{\Delta}(1)\delta q} \right] \tag{B.40}$$
$$= \frac{1}{2} \ln(\delta q) H(\tilde{x}(\kappa)) - \frac{B(\kappa)}{2\dot{\Delta}(1)\delta q},$$

where we have defined the quantities

$$\tilde{x}(\kappa) = -\frac{\kappa}{\sqrt{\Delta(1)-\Delta(0)}}, \tag{B.41a}$$

$$B(\kappa) = \kappa\sqrt{\Delta(1)-\Delta(0)}\,G\left(\tilde{x}(\kappa)\right) + \left(\kappa^2 + \Delta(1)-\Delta(0)\right)H\left(\tilde{x}(\kappa)\right). \tag{B.41b}$$

The free energy is

$$\phi = \frac{1}{2\delta q} + \frac{1}{2}\ln\delta q + \frac{\alpha}{2}\left(\ln(\delta q)H\left(\tilde{x}(\kappa)\right) - \frac{B(\kappa)}{\dot{\Delta}(1)\delta q}\right). \tag{B.42}$$

The derivative with respect to $\delta q$ is

$$2\frac{\partial\phi}{\partial\delta q} = \frac{1}{\delta q} - \frac{1}{\delta q^2} + \alpha\left(\frac{H\left(\tilde{x}(\kappa)\right)}{\delta q} + \frac{B(\kappa)}{\dot{\Delta}(1)\delta q^2}\right) = 0. \tag{B.43}$$

In the critical capacity limit, i.e. $\alpha = \alpha_c^{\text{RS}} - \delta\alpha$ we have that $\delta q$ scales linearly in $\delta\alpha$, $\delta q = C\delta\alpha$. We get

$$\begin{aligned}
2\frac{\partial\phi}{\partial\delta q} &= \frac{1}{C\delta\alpha} - \frac{1}{C^2\delta\alpha^2} + (\alpha_c - \delta\alpha)\left[\frac{H\left(\tilde{x}(\kappa)\right)}{C\delta\alpha} + \frac{B(\kappa)}{\dot{\Delta}(1)C^2\delta\alpha^2}\right]\\
&= \frac{1}{C\delta\alpha}\left[1 + \alpha_c H\left(\tilde{x}(\kappa)\right) - \frac{B(\kappa)}{C\dot{\Delta}(1)}\right] + \frac{1}{C^2\delta\alpha^2}\left[\alpha_c\frac{B(\kappa)}{\dot{\Delta}(1)} - 1\right] = 0.
\end{aligned} \tag{B.44}$$

The first term gives the scaling of $\delta q$, the second gives us the critical capacity in terms of the margin

$$\alpha_c^{\text{RS}} = \frac{\dot{\Delta}(1)}{B(\kappa)}. \tag{B.45}$$

Notice that imposing (B.45) is equivalent to impose that the divergence $1/\delta q$ in the entropy (B.42) is eliminated at the critical capacity (and the free energy correctly goes to $-\infty$ in that limit). In particular, in the zero margin case we get that $B(\kappa = 0) = \frac{\Delta(1)-\Delta(0)}{2}$ and therefore

$$\alpha_c^{\text{RS}} = \frac{2\Delta'(1)}{\Delta(1)-\Delta(0)} = \frac{2\int Dh\,\varphi'(h)^2}{\int Dh\,\varphi^2(h) - \left(\int Dh\,\varphi(h)\right)^2}, \tag{B.46}$$

as was previously derived in [25].

## B.5 1RSB Ansatz

### B.5.1 Entropy

In the 1RSB approximation and in the error counting loss case the entropy reads

$$\phi = \mathcal{G}_S + \alpha\mathcal{G}_E, \tag{B.47a}$$

$$\mathcal{G}_S = \frac{1}{2}\left(\frac{q_0}{1-q_1+m(q_1-q_0)} + \frac{m-1}{m}\ln(1-q_1) + \frac{1}{m}\ln(1-q_1+m(q_1-q_0))\right), \tag{B.47b}$$

$$\mathcal{G}_E = \frac{1}{m}\int Dz_0\,\ln\int Dz_1\,H^m\left(\frac{\kappa - \sqrt{\Delta(q_0)-\Delta(0)}\,z_0 - \sqrt{\Delta(q_1)-\Delta(q_0)}\,z_1}{\sqrt{\Delta(1)-\Delta(q_1)}}\right). \tag{B.47c}$$

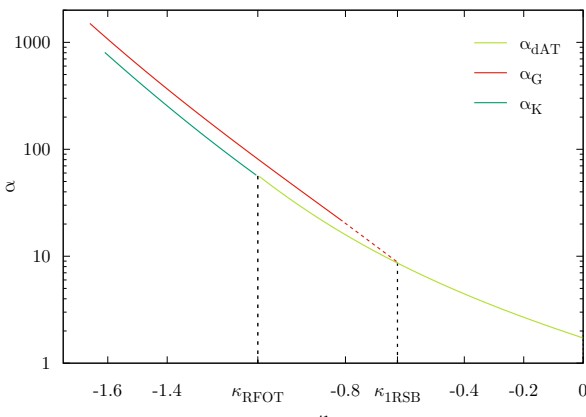

Figure 9: Plot of the dAT (eq. (B.37)), Gardner (eq. (B.48)) and Kautzmann transition lines as a function of $\kappa$ for the committee machine in the large width limit with the ReLU activation function.

### B.5.2 Gardner transition

In the 1RSB case usually the instability to the fRSB type of Ansatz (Gardner transition) develops at $q_1$. Imposing a 1RSB Ansatz in (39) and evaluating it in $q = q_1$ we get

$$
\frac{1}{(1-q_1)^2} = \frac{\alpha \ddot{\Delta}(q)}{\Delta(1)-\Delta(q_1)} \int Dz_0 \frac{\int Dz_1 H^m(\mathcal{A}(z_0,z_1)) GH^2(\mathcal{A}(z_0,z_1))}{\int Dz_1 H^m(\mathcal{A}(z_0,z_1))}
$$
$$
+ \frac{\alpha \dot{\Delta}^2(q)}{(\Delta(1)-\Delta(q_1))^2} \int Dz_0 \frac{\int Dz_1 H^m(\mathcal{A}(z_0,z_1)) \mathcal{W}^2(\mathcal{A}(z_0,z_1))}{\int Dz_1 H^m(\mathcal{A}(z_0,z_1))} ,
$$
(B.48)

where

$$
\mathcal{A}(z_0,z_1) \equiv \frac{\kappa + \sqrt{\Delta(q_0)-\Delta(0)}z_0 + \sqrt{\Delta(q_1)-\Delta(q_0)}z_1}{\sqrt{\Delta(1)-\Delta(q_1)}} .
$$
(B.49)

We plot the Gardner transition for the committee machine with the ReLU activation function in Figure 9.

### B.5.3 SAT/UNSAT transition in the 1RSB approximation

In order to compute the SAT/UNSAT transition in the 1RSB approximation, one needs to perform the limit $q_1 \to 1$ with $m = \tilde{m}(1-q_1) \to 0$ [12, 25, 29]. Therefore we express all in terms of $m$ by using $\delta q_1 = 1 - q_1 = \frac{m}{\tilde{m}}$ obtaining

$$
\phi = \frac{1}{2m} \left[ m \ln\left(\frac{m}{\tilde{m}}\right) + \ln(1-m+\tilde{m}(1-q_0)) + \frac{\tilde{m}q_0}{1-m+\tilde{m}(1-q_0)} + 2m\alpha \mathcal{G}_E \right] .
$$
(B.50)

In the limit $m \to 0$, we need to assure that the entropy goes to $-\infty$, so we need to impose that the coefficient of first order expansion of the free energy (which is of order $1/m$) vanishes. This is equivalent to impose that at the SAT/UNSAT transition

$$
\ln(1+\tilde{m}(1-q_0)) + \frac{\tilde{m}q_0}{1+\tilde{m}(1-q_0)} + 2\alpha_c \mathcal{F}(\kappa; q_0, \tilde{m}) = 0 ,
$$
(B.51)

or

$$
\alpha_c = \frac{\ln(1+\tilde{m}(1-q_0)) + \frac{\tilde{m}q_0}{1+\tilde{m}(1-q_0)}}{2\mathcal{F}(\kappa; q_0, \tilde{m})} ,
$$
(B.52)

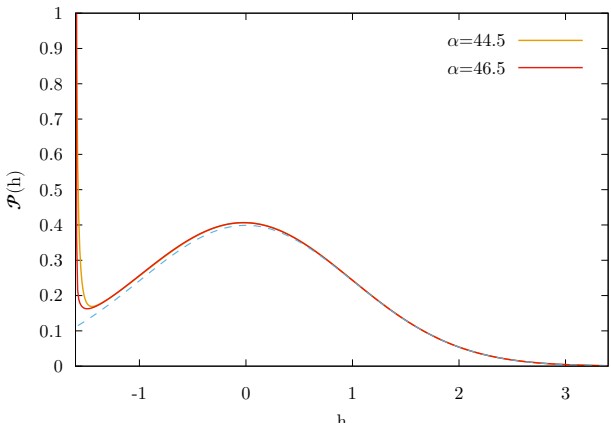

Figure 10: Stability distribution for $\kappa = -1.6$ and two values of $\alpha$. As $\alpha \to \alpha_c$ the distribution develops a power law behavior around small stabilities $h \sim \kappa$. We show in the dashed blue line a standard normal Gaussian distribution for comparison.

where

$$\mathcal{F}(\kappa; q_0, \tilde{m}) = \lim_{m \to 0} \int Dz_0 \ln \int Dz_1 H^m \left( \frac{\kappa - \sqrt{\Delta(q_0) - \Delta(0)} z_0 - \sqrt{\Delta(1) - \Delta(q_0)} z_1}{\sqrt{\dot{\Delta}(1) \frac{m}{\tilde{m}}}} \right). \quad \text{(B.53)}$$

Expanding the $H(x)$ function at large arguments $H(x) \simeq G(x)/x$ and performing the integral over $z_1$ one gets

$$\mathcal{F}(\kappa; q_0, \tilde{m}) = \int Dz_0 \ln \left[ H \left( \frac{\kappa + \sqrt{\Delta(q_0) - \Delta(0)} z_0}{\sqrt{\Delta(1) - \Delta(q_0)}} \right) \right. \quad \text{(B.54)}$$

$$\left. + \frac{\sqrt{\dot{\Delta}(1)} e^{-\frac{\tilde{m} \left( \kappa + \sqrt{\Delta(q_0) - \Delta(0)} z_0 \right)^2}{2(\dot{\Delta}(1) + (\Delta(1) - \Delta(q_0)) \tilde{m})}}}{\sqrt{\dot{\Delta}(1) + (\Delta(1) - \Delta(q_0)) \tilde{m}}} H \left( -\sqrt{\frac{\dot{\Delta}(1)}{\Delta(1) - \Delta(q_0)}} \frac{\kappa + \sqrt{\Delta(q_0) - \Delta(0)} z_0}{\sqrt{\dot{\Delta}(1) + (\Delta(1) - \Delta(q_0)) \tilde{m}}} \right) \right].$$

## C  Observables

Once the saddle point equations are solved, we can use the solutions not only to compute the entropy, but also other observables of interest.

### C.1  Distribution of stabilities

An observable of interest is the so called distribution of stability $\mathcal{P}(h)$, i.e.

$$\hat{\mathcal{P}}(h) \equiv \frac{1}{P} \sum_{\mu=1}^{P} \delta \left( h - \Delta^\mu(\boldsymbol{w}; \kappa) \right), \quad \text{(C.1a)}$$

$$\mathcal{P}(h) = \overline{\left\langle \hat{\mathcal{P}}(h) \right\rangle}, \quad \text{(C.1b)}$$

this quantity, also called "gap probability distribution" in the context of the jamming of hard spheres [6], quantifies in which fashion the constraints of the training set are satisfied. In the context of machine learning it has been recognized that well-generalizing solutions have a stability distribution that is small and flat around zero [22, 24]; those kind of solutions can be found by biasing the measure towards flat regions [25].

We can easily compute the partition function by rewriting the partition function in (5) as can be written as

$$Z = \int d\mu(\boldsymbol{w}) e^{-\beta \sum_\mu \ell(\Delta^\mu(\boldsymbol{w};\kappa))} = \int d\mu(\boldsymbol{w}) e^{P \int dh \hat{\mathcal{P}}(h)[-\beta \ell(h)]}. \tag{C.2}$$

The stability distribution can by taking a derivative of the free entropy with respect to the loss function, i.e.

$$\mathcal{P}(h) = -\frac{1}{\alpha\beta}\frac{\partial \phi}{\partial \ell(h)} = e^{-\beta \ell(h)} \int dz \, P(q_M,z) \mathcal{N}_{\Delta_1 - \Delta_{q_M}}(h+z+\kappa) e^{-f(q_M,z)}. \tag{C.3}$$

A generic observable $\mathcal{O}$ of the stability $h$, can be therefore easily expressed as an integral over the stability distribution

$$\begin{aligned}
\langle \mathcal{O} \rangle &\equiv \int dh \, \mathcal{P}(h) \, \mathcal{O}(h) = \int dh \, \mathcal{O}(h) e^{-\beta \ell(h)} \int dz \, P(q_M,z) \mathcal{N}_{\Delta(1)-\Delta(q_M)}(h+z+\kappa) e^{-f(q_M,z)} \\
&= \int dz \, P(q_M,z) e^{-f(q_M,z)} \int dh \, \mathcal{O}(h) e^{-\beta \ell(h)} \mathcal{N}_{\Delta(1)-\Delta(q_M)}(h+z+\kappa) \\
&= \int dz \, P(q_M,z) \frac{\int dh \, \mathcal{O}(h) e^{-\beta \ell(h)} \mathcal{N}_{\Delta(1)-\Delta(q_M)}(h+z+\kappa)}{\int dh \, e^{-\beta \ell(h)} \mathcal{N}_{\Delta(1)-\Delta(q_M)}(h+z+\kappa)}.
\end{aligned} \tag{C.4}$$

As an example, the fraction of violated constraints $z$ can be obtained by using the observable $\mathcal{O}(h) = \Theta(-h)$, i.e.

$$z = \int_{-\infty}^{0} dh \, \mathcal{P}(h). \tag{C.5}$$

## C.2 Pressure

The average stability of violated constraints or "pressure" [9] can be obtained by using as observable $\mathcal{O}(h) = -h\Theta(-h)$, which gives

$$p = -\int_{-\infty}^{0} dh \, \mathcal{P}(h) h = -\int dz \, P(q_M,z) \frac{\int_{-\infty}^{0} dh \, h \, e^{-\beta \ell(h)} \mathcal{N}_{\Delta(1)-\Delta(q_M)}(h+z+\kappa)}{\int dh \, e^{-\beta \ell(h)} \mathcal{N}_{\Delta(1)-\Delta(q_M)}(h+z+\kappa)}. \tag{C.6}$$

In the SAT phase, by definition $\mathcal{P}(h) = 0$ for $h < 0$, therefore the pressure (and also the fraction of violated constraints) vanishes. However one can study how it tends to zero with the temperature. For the number of error loss this decays exponentially to 0 with $\beta$ going to infinity. For the quadratic hinge loss it vanishes linearly to zero with $T = 1/\beta$; in the SAT region; indeed

$$p = -T \int dz \, P(q_M,z) \frac{\mathcal{N}_{\Delta(1)-\Delta(q_M)}(z+\kappa) \int_{-\infty}^{0} dh \, h \, e^{-\frac{h^2}{2}}}{\int_{0}^{\infty} dh \, \mathcal{N}_{\Delta(1)-\Delta(q_M)}(h+z+\kappa)} = -T \int dz \, P(q_M,z) f'(q_M,z). \tag{C.7}$$

Using the property

$$\frac{d}{dx} \int_{0}^{1} dx \, P(x,h) f'(x,h) = 0, \tag{C.8}$$

one gets

$$p = -T \int dz \, P(q_M,z) f'(q_M,z) = -T \int dz \, P(q_m,z) f'(q_m,z). \tag{C.9}$$

The "reduced pressure" presented in the main text is therefore related to the pressure by $p = T\tilde{p}$.

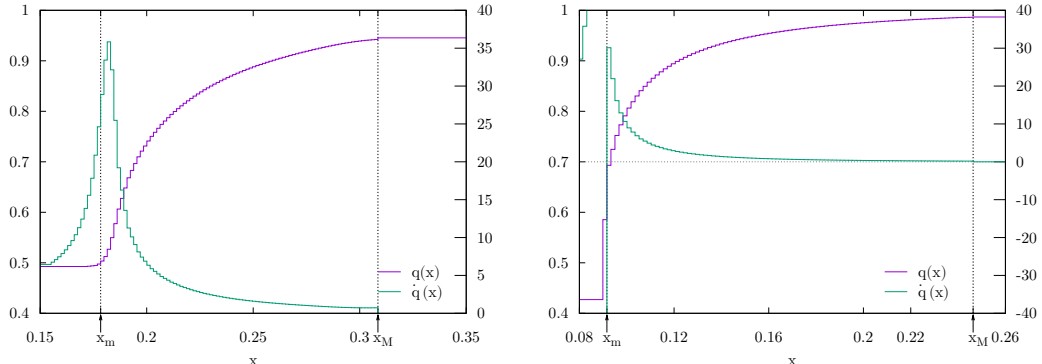

Figure 11: Behavior of $q(x)$ (violet) and $\dot{q}(x)$ (green) as a function of $x$ in the phase where typical states do not possess any gap (left panel, $\kappa = -1.27$ and $\alpha \simeq 18$) and a phase where they possess a gap (right panel, $\kappa = -1.4$ and $\alpha \simeq 26.7$). When there is no gap $\dot{q}$ is always positive in the range $x \in [x_m, x_M]$. A gap instead appears for a fixed $\kappa$ at a value of $\alpha = \alpha^{1+fRSB}(\kappa)$ where for $x \to x_m$, the denominator of (D.1) becomes zero, signalling an infinite derivative of $q(x)$. For $\alpha > \alpha^{1+fRSB}$, the denominator suddenly becomes negative at $x = x_m$.

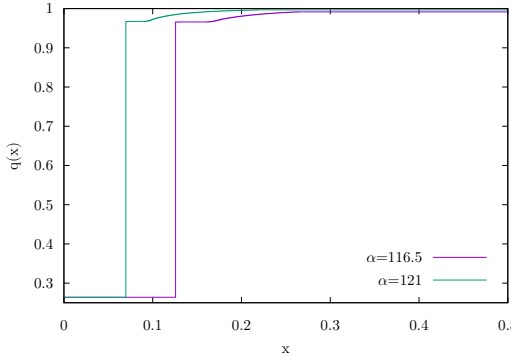

Figure 12: $q(x)$ deep in the Gardner phase (here $\kappa = -2.0$), where one can see clearly that the point $m$ where there is a jump is distinct with $x_m$.

# D Equation for $\dot{q}(x)$ and the transition to the overlap gapped phase

Higher order derivatives of the saddle point equation can give information to derivatives of $q(x)$ in the interval $[x_m, x_M]$. For example, deriving twice equation (39) and solving for $\frac{dq}{dx}$, we get

$$\frac{dq}{dx} = \frac{\frac{1}{\lambda^3(x)} + \alpha \dot{\Delta}^3 \int dh\, P(x,h) f''(x,h)^3}{\frac{\alpha}{2} \int dh\, P(x,h) \mathcal{B}(x,h) - \frac{3x^2}{\lambda^4(x)}}, \tag{D.1}$$

where

$$\begin{aligned}
\mathcal{B}(x,h) = {}& 6\dot{\Delta}^4 x^2 f''^4 + \dot{\Delta}^4 f''''^2 - 12\dot{\Delta}^4 x f'' f'''^2 + \dddot{\Delta} f'^2 \\
& + \left(3\ddot{\Delta}^2 + 4\dot{\Delta}\dddot{\Delta}\right) f''^2 + 6\ddot{\Delta}\dot{\Delta}^2\left(f'''^2 - 2x f''^3\right),
\end{aligned} \tag{D.2}$$

which in the case $\Delta(q) = q$ reduces to

$$\frac{dq}{dx} = \frac{\frac{1}{\lambda^3(x)} + \alpha \int dh\, P(x,h) f''(x,h)^3}{\frac{\alpha}{2} \int dh\, P(x,h)[6x^2 f''(x,h)^4 + f''''(x,h)^2 - 12x f''(x,h) f'''(x,h)^2] - \frac{3x^2}{\lambda^4(x)}} \,. \tag{D.3}$$

As we have described in the main text, we used equation (D.3) to evaluate the transition between the fRSB phase (no overlap gap phase), to the Gardner phase (which is overlap gapped). Indeed the transition is signalled by the divergence of the derivative of $q(x)$ at $x = x_m$, see Figure 11. If then one moves in a region $(\kappa, \alpha)$ deep in the Gardner phase (i.e. for $\kappa$ very negative and $\alpha$ large) one can see that the point where the $q(x)$ has a jump (i.e. for $x = m$) becomes visibly distinct and lower than $x_m$, see Figure 12. Similar transitions have been seen in [55], even if in a slightly different setting.

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
