# Peer review of "Exact full-RSB SAT/UNSAT transition in infinitely wide two-layer neural networks"

_SciPost Physics, doi:SciPost Phys. 18, 118 (2025)_

## Round 3 · Author Response

* * *
We report here the response to every single question of each referee. The references and the equations number we
cite here will refer to the new, corrected version of the paper.
* * *
#########################################################################
1. Report of the first referee
#########################################################################
The authors investigate two continuous non-convex constraint satisfaction problems: the negative-margin percep-
tron and the infinite-width tree-committee machine. They perform a static analysis in the full Replica Symmetry
Breaking (fRSB) regime, computing the SAT-UNSAT transition line for these models.
For the negative perceptron, they identify a Gardner transition, where the typical solutions develop an "overlap
gap." This feature implies that Approximate Message Passing (AMP) algorithms aren’t guaranteed to converge in
this regime.
Additionally, the authors provide numerical evidence that Gradient Descent (GD) fails to find solutions before
the models reach capacity, irrespective of the overlap gap. This observation suggests the presence of an algorithmic
threshold that separates the theoretical capacity from practical solvability with GD.
The computation of the transition lines in the fRSB phase is an important technical result, as it requires a non-trivial
numerical approach. The implications of the phase diagram for AMP algorithms are very interesting and establish
connections with the optimization literature. The GD data suggesting the presence of an algorithmic gap is also very
interesting. However, I think it would benefit from an expanded discussion to provide additional details and insights.
Here are some questions on Section 6:
- How many runs were performed to estimate the probability of finding solutions with GD?
We did as many runs as we could in order to have error bars (not visualized for clarity) that are the
smallest as possible. Depending on the size N this corresponds to a number between 10 up to 100
runs.
* * *
We did as many runs as we could in order to have error bars (not visualized for clarity) that are the
smallest as possible. Depending on the size N this corresponds to a number between 10 up to 100
runs.
* * *
- How robust are these probability curves to changing GD hyperparameters, such as the maximum number of
training epochs? From the left panel of Fig. 6, it is unclear whether fine-tuning the hyperparameters could enable
GD to reach αc .
* * *
Our analysis revealed that the number of training epochs is indeed a critical hyperparameter. We
selected as many epochs as possible in order to maintain computational feasibility and at the same
time to ensure proper exploration of the model’s learning capacity. The learning rate was chosen as
a balanced compromise to ensure stable convergence without overshooting or getting trapped in local
minima. Specifically, while decreasing the learning rate could actually help the convergence of the
algorithm, this comes at a significant computational cost, since the overall number of epochs needed
to reach a solution, and so the average learning time, will dramatically increase as well.
* * *
- Is there a way to compute the algorithmic threshold for GD? The introduction mentions that the methods
applicable to discrete weights do not generalize to the continuous case; is there an alternative approach?
* * *
We are not aware of a method to compute the algorithmic threshold of GD. The method in reference
[20] are able to predict the constraint density αhard where all algorithms should fail to find solutions
in polynomial time. This is obtained by studying where the flattest cluster of solutions suddenly
breaks apart. Those methods are therefore not able to predict the algorithmic threshold of GD which
usually stops working before αhard . The same methods exposed in [20] however do not seem to work
for continuous weight models. We are currently not aware of a solution to this dilemma. This work
aimed at exactly going in the direction to understand if such an algorithmic threshold for GD exists, comparing it with an exact calculation of the SAT/UNSAT transition.
* * *
-Is there a notion of Overlap Gap for the atypical solutions found by GD?
* * *
It is possible to define it, if one could theoretically describe what is the asymptotic probability dis-
tribution of the weights that the GD algorithm targets with a given fixed learning rate. The authors
are not aware of any results in this respect. If this probability distribution is accessible, one could in
principle calculate an ‘overlap gap’ transition threshold for GD and conjecture that it is equal to the
algorithmic threshold of GD.
* * *
Here are some typos:
- pag. 3, line 1: missing reference
- pag. 4, eq. (3): it’s l ∈ [K]
- pag. 7, eq. (15b): it’s λb instead of λbl
- pag. 27, before eq. (83): it’s "energetic term" instead of "entropic term"
- pag. 31, after eq. (103): it’s "Fig. 8" instead of "Fig. 11"
* * *
All the typos and mistakes have been corrected. Thanks.
* * *
#########################################################################
2. Report of the second referee
#########################################################################
This work considers the problem of how many Gaussian random patterns ξ µ ∼ N(0, IN ), µ ∈ [P ] a two-layer
neural network can asymptotically classify within a margin κ ∈ R when the corresponding binary labels are random
y µ ∼ Rad(1/2) (a.k.a. the storage capacity problem). The analysis focus on a particular two-layer architecture with
fixed second layer weights and K hidden-units that partition the input data ξ µ in K disjoint patches ξlµ of size N/K,
a.k.a. as the tree committee machine:
[...]
Previous literature on this problem has mostly focused in the RS and 1-RSB approximations of the Gardner volume.
The main contribution of this work is to carry out a full-RSB analysis, and to precisely characterize the transitions
between the different levels of RSB as a function of the sample complexity α = P/N and margin κ. This allows to
exactly compute the capacity threshold αc (κ) of maximum number of random patterns the network can correctly
classify.
Comments, questions and suggestions
1. Page 3 in the Introduction:
For continuous weight models instead, the picture is not as clear: the same tools used for binary models provide
an algorithmic threshold that can be easily overcomed by simple algorithms [19].
I understand what you mean, but I would not call an ”algorithmic threshold” a threshold which can be “easily
overcomed”.
* * *
It has been corrected by eliminating the word “algorithmic”.
* * *
2. Bottom of Page 5:
Interestingly, this algorithm can be proven to reach capacity, provided that the typical states exhibit no overlap
gap, i.e. the overlap distribution of typical states is with a compact support.
* * *
Thanks for pointing it out. The term “compact” was incorrect, we have substituted it with
“connected”.
* * *
3. Can you further elaborate on the difference between nOG and OGP? This is not very clear from the brief
discussion in the end of Page 5. Giving an explicit example of a problem / overlap distribution which has nOG
but no OGP would be useful.
* * *
We have better clarified the difference between the OG condition and the OG property in section
2.1. The difference is that while the OGP refers to worst case conditions, i.e. there exist no
two solutions with overlaps in a certain interval (q1 , q2 ), the OG condition only takes into account
typical solutions. This second property can be easily inferred by looking at the distribution P (q):
if its support is an interval then the nOG condition holds, if not the OG condition holds.
* * *
4. The central limit theorem argument in Section 3.2 is a key part in the derivation, but I feel some important
points are not properly discussed. For instance
• You take the large-width K → ∞ limit before the high-dimensional limit N, P → ∞ at fixed α = P/N .
Unless these two limits commute, this means you effectively assume K grows faster than N . Is it clear
these two limits commute here? In either case, this should be stressed both in this section and before,
stating that all your capacity results hold strictly for the infinite width case.
* * *
Thanks for pointing this out, indeed it is not clear from the way the calculations are reported.
In reality, we are performing the limit K → ∞ after the limits N, P → ∞.When we calculate
the energetic term in equation (14c) we are implicitly assuming that we will calculate it at
the saddle point. Only then do we consider the large K limit of this term. We have now
clarified this in section 3.2. Computing the entropy at a fixed value of α = P/N = O(1) and
αK = K/N = O(1) with K, P, N all jointly going to infinity is an open problem.
* * *
• How important is the 1/ K scaling in the second-layer for this argument? This seems to be closely related
to the fact you get a NNGP in the following. I understand that ŷ is invariant under this scaling due to the
sign, but the stability ∆µ upon which depends the loss function is not. For instance, would the argument
hold in a “mean-field” scaling 1/K?
* * *
In our paper we are employing what is called the “NTK” parametrization, so our scaling of
the output by 1/ K is crucial. In this parameterization, however we study a different scaling
regime with respect to the one corresponding to the classic NTK parameterization papers,
see (arXiv:1806.07572). There the input dimension N , and the number of samples P are fixed
to finite values while the width K is sent to infinity; here instead P and N are large (with
their ratio fixed), and K is large but small compared to them.
Scaling the output 1/K as in the “mean field” parameterization or in the µP (i.e. the “maxi-
mum update parameterization”), would lead to a completely different computation from the
very start, if one does not want to get a meaningless result (indeed scaling the output simply
by 1/K in our framework, one gets that only the mean of the activation function contributes
in equation 18 and there will be no dependence on the overlap).
* * *
• Can you be be more explicit in the passage from the first to the second equality in eq. (18)? I didn’t find
this in the Appendix, and would encourage you to detail it better there.
* * *
We thank the referee for the suggestion. We have directly stated the result in equation 18 now and created a subsection in the appendix (see Appendix B.2.1), with a careful derivation of the result. Appendix B.2.1 has also been referenced in the main text, just before equation 18 itself.
* * *
5. From your argument, it looks like the critical storage capacity αc is independent of the width K (differently
from the fully-connected committee machine). Is this right?
* * *
Yes, if one defines the constraint density as the fraction of patterns to the input dimension.
However in some literature it is defined as the fraction of patterns divided by the number of
weights. Notice that the first and the second definition is actually the same in the tree committee
machine model, as the number of weights is equal to the input dimension. However the two
definitions are different in the fully connected committee machine, as there one has N K learning
parameters, and one would always get a diverging capacity in the first case, but one could obtain
a finite result by using the second definition. Indeed as already pointed out in ref. [24] which
studied the critical capacity of the generic activation function tree committee machine in the RS
and 1RSB, one obtains a non-trivial diverging capacity with K for certain activation function
like the sign as was already observed in papers from the ’90. The nature of this divergence was
studied more in depth in ref. [13]. Recently the results outlined in [24] have been extended to
the fully connected case in Nishiyama, Ohzeki, Journal of the Physical Society of Japan (2025),
which we have added as reference in the paper.
* * *
6. You refer to the “Gardner transition” several times in the text, but this is never explicitly defined.
* * *
We explicitly clarified in section 3.4 that by Gardner transition we mean a transition from a
stable 1RSB to a Gardner phase which is defined in section 5.1.
* * *
7. Last sentence of the Conclusion:
Finally, we compared our estimates of the SAT-UNSAT threshold with the performance of Gradient Descent.
In all cases analyzed we have given evidence that Gradient Descent stops finding solutions before the exact
SAT/UNSAT threshold that we computed, thus implying the presence of an algorithmic gap.
The wording of this sentence is too strong - the numerical evidence does not “imply” but “suggest” the presence
of an algorithmic gap.
* * *
Corrected as suggested.
* * *
8. Bottom of Page 17:
When training a tree committee machine we have empirically observed that the first method leads to a larger
probability of finding a solution, while for the negative perceptron the second method works best.
This is curious. Do you have an intuition why? In particular, I am not sure the second method really makes
sense in the context of the analysis. Indeed, the unconstrained risk function will be different from the constrained
one — why would you expect the same algorithmic threshold?
* * *
The second method is particularly suited to the perceptron as it is a linear model, and thus
renormalizing the weight vector simply affects the scale of the output. Indeed in ref [37] the
authors prove that the second method is guaranteed to find a κ margin solution. However this
doesn’t hold for two layer machines, as you point out, since renormalizing the independent neu-
rons fundamentally changes the output. So perhaps it is not surprising that renormalizing the
neurons at every epoch is the best choice for the tree-committee machine.
* * *
9. Overall, I think the paper lacks a clear “take home message”. The authors start by motivating that the current
RS and 1RSB results provide only an upper bound of the storage capacity of these networks. But what do we
learn from correcting a 10^{-2} digit in this constant (in the worst case discussed)? For instance, it would be nice
to have a discussion on how the activation function impact the capacity. From Table 1, it seems it does not
change much — is this always the case or one can build examples with larger gaps?
* * *
Indeed the corrections to previous estimates to the critical capacity are small. But of course this
could have been only verified a posteriori: before computing it we had no clue that the value
of the actual capacity would have been so near to the 1RSB computation. Moreover without
actually computing it exactly, one could not argue about the presence of an algorithmic hardness
phase for gradient descent or any other algorithm. Unfortunately, it is not completely obvious
to tell if the RSB corrections will be small or large, just by looking at the shape of the activation
function, but one needs to solve each time the full-RSB equations that we derived. Finally, we
underline that our method in turn gives us full access to the distribution P (q), from which one
can actually implement optimized learning algorithms based on AMP, that, as shown in ref. [27]
are proved to work up to capacity at least in the no OG condition phase.
* * *
Small typos and suggestions
• Broken reference [?] in the top of Page 3.
• There is a φ too much between eqs. (1) and (2). From the calculation that follows, I think you meant no φ in
eq. (2).
• For the sake of clarity, state that throughout the manuscript you assume that N/K is an integer.
• The notation Θ for the Heaviside step function in eq. (6) is standard in Physics, but not in CS. Since this
work can also interest people in this community, I suggest the authors to define it explicitly or say it in words
in/around eq. (6).
• Page 8, “semidefite” → “semidefinite”.
• Sometimes you write “tree committee machine”, sometimes “tree-committee machine” and sometimes “tree
committee-machine”.
* * *
All the typos have been corrected and the suggestions implemented. Thanks.
* * *
#########################################################################
3. Report of the third referee
#########################################################################
This paper conducts a detailed analysis of the phase diagram of the tree committee machine, a model of two-layer
neural networks which includes the perceptron as a special case. The paper considers the case where the weights of
the first layer are on the sphere and the weights of the second layer are fixed, and derives via the replica method the
(free) entropy of the set of solutions (‘interpolators’) when there are P = αN random Gaussian patterns in dimension
N in the large system limit under a general full RSB ansatz, and provides an accurate estimate of the SAT/UNSAT
transition for various popular activation functions.
The paper then considers the special case of the negative perceptron. The main contribution of the paper is to
unveil the existence of a Gardner phase bordering the satisfiability threshold for a large unbounded range of negative
values of the margin κ. In this regime the two-replica overlap distribution exhibits an atom at its left-end followed
by a gap followed by a continuous part, a possibility which was not considered in previous literature where only the
continuous part was established/assumed to exist.
The existence of an ‘overlap gap’ in this Gardner phase has an important algorithmic consequence. The incremental
approximate message passing algorithm (IAMP) of ref. [26] is known to succeed at returning a solution when there
is no gap in the overlap distribution. The authors show that the ‘no overlap gap’ (FRSB) phase is a bounded region
bordering the SAT/UNSAT transition for relatively moderate values of the negative margin κ, followed by the Gardner
phase where this algorithm must fails. In particular the IAMP algorithm does not succeed up to satisfiability for all
κ < 0.
The authors also conduct a numerical experiment with gradient descent with two loss functions (the quadratic hinge
loss and the cross entropy) and show that this algorithm fails much below the SAT/UNSAT threshold.
This work unveils a new phenomenon in the negative perceptron with important consequences regarding the computational tractability of finding solutions in toy models of neural networks. I have two comments:
1. I find the shape of the boundary between the fRSB phase and the Gardner phase (the dotted blue line α1+fRSB
in Figure 5) particularly intriguing for the following reason: let’s say that a point in the (κ, α) plane is solvable if
there exists an efficient algorithm for returning a solution to the perceptron problem at those parameters. Then
any point to the bottom-left of a solvable point is also solvable since one can add constraints and/or make the
margin tighter, solve the harder problem then return the solution found. (This is a simple argument justifying
the fact that the SAT/UNSAT transition line is decreasing. Now, if the boundary κ → α1+fRSB (κ) is increasing
as depicted in Figure 5, then there must exist solvable points in the Gardner and 1RSB phases of the model.
It is not clear that this is impossible but it would be interesting since this would imply an absence of (worst
case) overlap gap in a 1RSB phase. It would be instructive if the authors could provide more details on the
estimation of this line.
* * *
We thank the referee for his/her comment; what you are pointing out was not properly discussed
in the text. Your argument indeed shows that there are parts of the 1RSB and Gardner phase
which are solvable. This is not a contradiction, as arguments about the existence of an algo-
rithmic hard phase are always related to worst case overlap gap (which we call “Overlap Gap
Property” in the paper), while what holds in the 1RSB and Gardner phases is a “typical” overlap
gap (which we call “Overlap Gap Condition”). Previous works (e.g. ref [20, 26, 40]) have shown
that, it is sometimes possible to find solutions even in phases where the Overlap Gap Condition
holds, by looking for atypical solutions. What we can say for sure is the following: denote by
(κ_∗ , α_∗ ) the point in which the α^{1+f RSB} (κ) line intersects the critical capacity. Then for all κ < κ∗
and α < α∗ the system is solvable. For greater values of α we cannot say at the moment. We have
added a comment on this in section 5.2.
The numerical criterion we used for the estimation of this line is described in section 5.2. For a
set of values of κ, we gradually increase α and look at the q̇(x). The first value of α for which this
becomes negative is our estimate of α^{1+f RSB} (κ).
* * *
2. My second comment is about the gradient descent experiments. The success probability should transition
sharply from 1 to 0 at the some threshold αGD for large system size. The transition does appear to be sharp in
Figure 7 but not so much in Figure 6. One run appears to find solutions beyond the critical threshold αc , which
means finite size effects are still present. Perhaps the authors could run larger instances and average more runs
so all curves are nearly vertical and collapse on top of each other. Moreover, ref. [36] has a rigorous analysis of a
linear programming algorithm and experimental results on some version of gradient descent. Perhaps depicting
their transition line in the same phase diagram would also be instructive for the sake of comparison.
* * *
We have added simulations for larger system sizes in the case κ = −1.5. For κ = −0.5 we are
close to the limit, as we are already considering very large system sizes. Let us comment that
in Figure 6 the transition doesn’t appear as sharp as we are zooming the plot in a region that is
very close to the critical capacity, and the scale in α is small. If one plots both of the figures on
a larger α scale of course the curves that we have look very sharp. Even with the zoom near the
critical capacity, our plots (also with the new data added for κ = −1.5) however show that as N
increases, the transition becomes sharper, thus indicating that the actual algorithmic threshold
can be estimated around the point where the larger size curves approach a 0.5 probability of
success.
As for the thresholds from ref. [37], the only case we analyze both is κ = −1.5, so we decided
to keep the plot as they are. Moreover, their threshold for the linear programming algorithm is
at much lower values of α (∼ 15 which actually falls in the RS phase αdAT ≃ 18.74), so it does not fit in the α range of our plot. Their transition for gradient descent instead is comparable to ours, with the only difference that we consider
also larger system sizes. We have produced an equivalent of the figure 6 of the paper with the same scale of ref. [37]
together with the linear programming threshold, but did not find a way to upload it here in the response. Sorry about that.
* * *
I support acceptance of the paper conditional on a revision addressing the previous points.

---

## Round 3 · List of Changes

• We have modified section 2.1 to better clarify the difference between the “OG condition” and
OGP.
• We have clarified the order of the limits N, K → ∞ in section 3.2.
• We added section B.2.1 in the appendix with a full derivation of equation 18 of the main text.
• We clarified in section 3.4 the meaning of the “Gardner transition”.
• We have expanded section 5.2 according to the suggestion of referee 3.
• We have performed larger scale simulations for the negative margin perceptron and updated
Figure 6 (right) accordingly.
• All the typos, mistakes and incorrect wording pointed out by the referees have been fixed.

---

## Editorial Decision

published